# LMRL Gym: Benchmarks for Multi-Turn Reinforcement Learning with Language Models

**Marwa Abdulhai** [1]  **Isadora White** [1]  **Charlie Snell** [1]  **Charles Sun** [1]
**Joey Hong** [1]  **Yuexiang Zhai** [1]  **Kelvin Xu** [2]  **Sergey Levine** [1]

## Abstract

Large language models (LLMs) provide excellent text-generation capabilities, but standard prompting and generation methods generally do not lead to intentional or goal-directed agents and might necessitate considerable prompt tuning. Even the best current LLMs rarely ask clarifying questions, engage in explicit information gathering, or take actions that lead to better decisions after multiple turns. Reinforcement learning has the potential to leverage the powerful modeling capabilities of LLMs, as well as their internal representation of textual interactions, to create capable goal-directed language agents. This can enable intentional and temporally extended interactions, such as with humans, the emergence of complex skills such as persuasion, and long-horizon strategic behavior, such as in the context of games. Enabling this requires the community to develop reliable reinforcement learning algorithms for training LLMs. Developing such algorithms requires tasks that can gauge progress on algorithm design, provide accessible and reproducible evaluations for multi-turn interactions, and cover a range of task properties and challenges in improving reinforcement learning algorithms. Our paper introduces the LMRL-Gym benchmark for evaluating multi-turn RL for LLMs, together with an open-source research framework for getting started on multi-turn RL with offline value-based and online policy-based RL methods. Our benchmark consists of 3 Interactive Dialogue tasks and 5 RL Capability tests for a total of 8 tasks, which require multiple rounds of language interaction and cover tasks in open-ended dialogue and text games.

[1]University of California, Berkeley [2]Google. Correspondence to: Marwa Abdulhai <marwa_abdulhai@berkeley.edu>.

*Proceedings of the 42nd International Conference on Machine Learning*, Vancouver, Canada. PMLR 267, 2025. Copyright 2025 by the author(s).

## 1. Introduction

Large language models (LLMs) have demonstrated remarkable abilities when naturally conversing with humans (OpenAI, 2023; 2022; Touvron et al., 2023; Google, 2023), answering questions and responding to requests (Shuster et al., 2022b;a; Qin et al., 2023), and even performing coding tasks (Chen et al., 2021b; Wang et al., 2023b). Many of these capabilities are enabled by learning to emulate humans from large datasets of text from the web (Völske et al., 2017; Shuster et al., 2022a; Yao et al., 2023), learning from examples "in context" (Brown et al., 2020), as well as learning from other sources of supervision such as instruction datasets (Mishra et al., 2022; Wei et al., 2022; Wang et al., 2022b) and preference fine-tuning with RLHF (Ziegler et al., 2020; Ouyang et al., 2022). However, directly applying LLMs in settings that require planning or multi-turn interactions presents new challenges. LLMs are not explicitly goal-directed, as they are not optimized to directly solve particular tasks, but rather to produce text that resembles the distribution of human-provided examples or accords with human preferences (Ziegler et al., 2020; Stiennon et al., 2020; Wu et al., 2021; Bai et al., 2022a). This challenge is apparent in solving temporally extended tasks, such as multi-turn dialogue (Irvine et al., 2023; , FAIR), complex tool use (Wang et al., 2022a), multi-step games (Hendrycks et al., 2021b), and other interactive applications. In principle, LLMs should contain the knowledge necessary to succeed in such settings: if the multi-turn interactions center around problem domains that are well represented in the model's training data (such as dialogue), well-trained LLMs should already serve as powerful predictive models in such settings. However, leveraging this predictive knowledge to derive effective actions and strategies requires not just emulating humans, but also planning and optimization.

Multi-turn reinforcement learning (RL) (Sutton & Barto, 2018) in principle offers a path to enable LLMs to do just that. RL could enable goal-directed reasoning and planning in interactive multi-turn settings, including complex dialogue, games, and tool use. We hypothesize that RL could serve as a powerful tool for LLM training, not only for training models to accord with human preferences, but

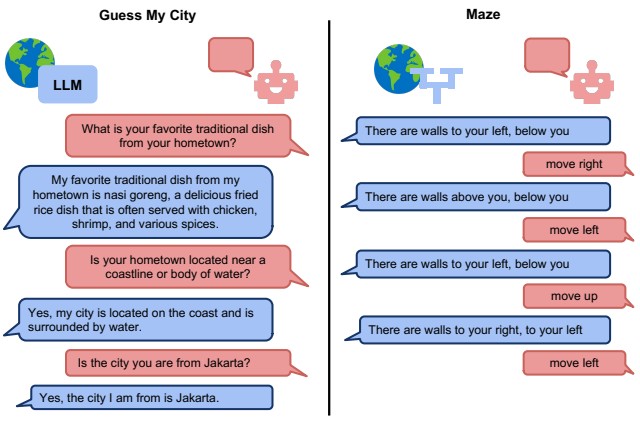

*Figure 1.* **Overview of LMRL-Gym:** We show sample trajectories from two tasks in our benchmark. In the Guess My City task, the agent learns to ask questions to guess the city the oracle is from, while in the Maze task, the agent learns to make the correct moves based on cues from the oracle.

more generally to accomplish tasks in an intentional and goal-directed manner. Text generation can be viewed as a sequential decision-making process, treating a sequence of tokens as a trajectory. Many tasks, such as successfully answering questions or eliciting a desired reaction from a user, can then be framed as optimizing some reward function over these trajectories. However, despite extensive interest in RL for LLMs in recent years, much (though not all) of the recent research in this area has focused on "single-step" RL problems, where a single response is optimized for some quality metric, typically derived from human preference signals (Stiennon et al., 2020; Ziegler et al., 2020; Ouyang et al., 2022; Bai et al., 2022a; Anthropic, 2023; Ramamurthy et al., 2023; Christiano et al., 2023; Casper et al., 2023).

While some works have sought to apply RL for multi-turn tasks (Singh et al., 1999; Li et al., 2016; Shah et al., 2016; Kwan et al., 2022), particularly for goal-directed dialogue (Lewis et al., 2017; Verma et al., 2022), there has been comparatively little research on improving the underlying RL algorithms and very little head-to-head comparison on same sets of tasks. This is perhaps unsurprising: it is easier to evaluate improvements to algorithms for single-turn text generation as compared to multi-turn generation. Multi-turn dialogue requires an interactive evaluation procedure rather than just a static dataset. There is no established protocol for such evaluations, and the "gold standard" constitutes costly and time-consuming studies with human participants.

In this work, we aim to address this challenge and make it possible for RL algorithm researchers to iterate on developing better RL methods for multi-turn language-based interaction tasks, such as dialogue and games. We posit that benchmarking RL algorithms for LLMs presents a very different set of challenges and merits a different set of so-

lutions compared to other benchmarks in NLP. While most NLP benchmarks are based on standard supervised machine learning paradigms, with a training set and a test set (Marcus et al., 1993; Tjong Kim Sang & De Meulder, 2003; Socher et al., 2013; Rajpurkar et al., 2016; Wang et al., 2019; Williams et al., 2018), RL benchmarks require simulators that the trained agents can interact with to measure their performance. In this paper, we use an LLM to simulate a conversation partner in dialogue tasks. While the behavior of the LLM may deviate from human behavior, we verify in a human study in Appendix A that our LLM simulators produce natural text reflecting human norms of conversation. However, our goal is *not* to utilize this approach to benchmark whether LLMs are *good at talking to humans*, but rather as a way to test RL algorithms with datasets that are sufficiently difficult and complex to gauge how effective they might be if they were then trained on data from real humans. Specifically, our benchmark aims to rigorously stress-test the ability of RL algorithms to enable complex goal-directed behaviors in LLMs. To this end, LMRL-Gym also includes a set of text-based strategy games, in addition to the dialogue tasks, that are aimed at providing a more controlled and focused diagnostic assessment of specific RL capabilities.

Our proposed benchmark, LMRL-Gym, consists of 8 tasks. Three tasks are Interactive Dialogue tasks designed to simulate real-world interactions with humans, requiring information gathering (20 Questions, Guess My City) and negotiation (Car Dealer). Five tasks are RL Capability Tests, which are text games designed to isolate specific capabilities of RL training. Each task comes with an offline dataset that can be used for offline RL training, and a "simulator" that can be used to evaluate the performance of the agents in multi-turn interactive tasks. We provide a research framework and toolkit for researchers and practitioners to get started with multi-turn RL for LLMs. This framework includes implementations of PPO (Schulman et al., 2017), ILQL (Snell et al., 2022a), and several baseline methods, implemented in an extensible way designed for future development of tasks, experimentation, and algorithm design.

## 2. Related Works

**Datasets, benchmarks, and libraries.** Benchmarks and datasets have been an important factor for driving progress in NLP in domains that include machine translation (Tiedemann, 2012; Bojar et al., 2016), natural language understanding (Rajpurkar et al., 2016; Wang et al., 2019; Hendrycks et al., 2020; 2021a; Ramamurthy et al., 2023), and solving math problems (Cobbe et al., 2021). However, these tasks generally do not involve multi-turn interaction and do not come with rewards, making them hard to adapt to RL research. For example, the standard for eval-

uating dialogue agents has been to run a human subjects study, but this is time-consuming and costly. Some works have proposed text games for evaluating language-based agents (Chevalier-Boisvert et al., 2018; Hausknecht et al., 2019; Yuan et al., 2019; Fan et al., 2020; Hausknecht et al., 2020; Guo et al., 2020; Ammanabrolu et al., 2020; Yao et al., 2020; Hendrycks et al., 2021b; Singh et al., 2021; Wang et al., 2022a; Yao et al., 2022; Jansen & Côté, 2022; Yao et al., 2023; Zhang et al., 2023; Gontier et al., 2023) and interactive dialogue (De Bruyn et al., 2022b;a). Our aim is to cover a variety of problem settings that reflect challenges in open-vocabulary interaction in addition to text games, that also specifically evaluate offline RL capabilities, which is not done by prior works. Motivated by successes in using LLMs to generate synthetic data (Hausknecht et al., 2019; Park et al., 2023; Bai et al., 2022b), our proposed tasks are based on synthetic data. While such data may differ from natural text, the scope of our benchmark is specific to evaluating RL algorithms, not the ability to interact with humans.

**RL for language models.** RL for language models has seen success in aligning LLMs with human preferences (RLHF) (Ziegler et al., 2020; Stiennon et al., 2020; Bai et al., 2022a;b; Ouyang et al., 2022; Christiano et al., 2023), optimizing non-differentiable objectives for machine translation (Wu et al., 2016; Nguyen et al., 2017; Kiegeland & Kreutzer, 2021), generation (Tambwekar et al., 2019; Pang & He, 2021; Pyatkin et al., 2022), dialogue (Cuayáhuitl et al., 2015; Georgila & Traum, 2011; Li et al., 2016), question answering (Pyatkin et al., 2022), and summarization (Paulus et al., 2017; Böhm et al., 2019; Wu & Hu, 2018). These include RL methods that learn by directly interacting with the environment (online RL) (Carta et al., 2023) and RL methods that only use a static dataset (offline RL) (Jaques et al., 2020; Snell et al., 2022a; Jang et al., 2022; Verma et al., 2022; , FAIR). However, many of these works operate in the singe-step bandit setting, and do not consider multi-turn goal-directed tasks. Our benchmark, on the other hand, focuses on tasks involving multiple turns of interaction with clearly defined goal-based reward functions.

**Capabilities of LLMs.** There has been a surge in the capabilities of LLMs for generation (Ghazvininejad et al., 2017; Radford et al., 2019), dialogue (Lewis et al., 2017; Jaques et al., 2017; Shuster et al., 2022b; Snell et al., 2022b), question answering (Pyatkin et al., 2022), summarization (Paulus et al., 2017; Böhm et al., 2019; Wu & Hu, 2018), text-based games (Narasimhan et al., 2015; Hausknecht et al., 2019), translation (Gu et al., 2017), and more. However, these are often supervised learning tasks that do not test the LLMs' abilities to achieve a specific long-term objective. Research on dialogue generation (Jaques et al., 2017; He et al., 2018; Shuster et al., 2022b;a) has often focused on generating

feasible-looking agent dialogue without explicit consideration for some multi-turn objective. Our benchmarks allow for the development of algorithms that enable LLMs to *interact* with an environment to achieve long-term objectives, by providing tasks with online simulators and offline datasets.

## 3. Multi-Turn Generation with RL and Language Models

This section introduces the conceptual foundations of using reinforcement learning for multi-turn generation with language models. We introduce a definition of the Markov decision process for language and a framework for the methods we focus on in this paper.

**Definitions.** We formalize language generation tasks as a partially observable Markov decision process. We define the state to be the history of tokens and an action as the next token generated by the model. An observation is a single token $o_i$ in the history. The probability of generating the next token is dependent on all of the previous observation tokens $o_i$. Therefore the Markovian state $s$ is formed by the concatenation of all the previous tokens $[o_0, \ldots, o_i]$. A policy $\pi$ defines the agent's behavior by taking in the current state $s$ and outputting a new action token $a$ to get $s_{i+1}$. The environment assigns a reward $r(s, a)$ based on the entire sequence of tokens so far. The tokens in the state are either generated by the policy $\pi$ or the environment. For example, in the Car Dealer task, the policy generates the tokens for the Seller's utterance and the environment generates the tokens for the Buyer. The full history of their conversation would form the state. A complete sequence of tokens is referred to as a trajectory $\tau = o_0, \ldots, o_T$. The goal of RL is to produce a policy $\pi^*$ that maximizes the expected discounted sum of rewards over trajectories ($\tau$) under the policy $\pi^* = \arg\max_\pi \mathbb{E}_{\tau \sim \pi} \left[ \sum_{t=0}^{T-1} \gamma^t r_t(s_t, a_t) \right]$, where $\tau$ represents the trajectory.

**RL Algorithms.** Several possible RL algorithms could be used to train language models for multi-turn tasks (Jaques et al., 2020; Verma et al., 2022; Snell et al., 2022a; Schulman et al., 2017; Stiennon et al., 2022; Bai et al., 2022a; Casper et al., 2023). Policy gradient methods, such as PPO (Schulman et al., 2017), directly compute the gradient of the RL objective with respect to the model parameters. Value-based methods estimate a state-action ($Q$) and/or state-value ($V$) function. The state-action or state-value function forms a policy by either 1) acting greedily with respect to the Q-function or 2) perturbing the base model's logits with the learned action-value functions (Snell et al., 2022a). RL methods for training LLMs can be *online* or *offline*. Online methods repeatedly interact with the environment, collecting additional data during training. Offline RL instead learns

to extract the best behaviors from an existing, potentially suboptimal dataset. Due to the large amount of existing text interactions on the internet, offline RL is an ideal setting for training language models. Therefore, our work primarily focuses on benchmarking offline RL algorithms. However, our tasks also fully support online RL and we include an online PPO baseline in our evaluation.

## 4. The LMRL-Gym: Synthetic Benchmarks for RL with Language

Our benchmark consists of 8 tasks grouped into two categories: RL Capability tasks and Interactive Dialogue tasks. The RL Capability tasks focus on desirable capabilities for RL algorithms for LLMs such as strategic decision-making, credit assignment, trajectory stitching, partial observability, and use of complex language. For the interactive dialogue tasks, we model them after real-world interactions with humans, such as persuading someone to buy a car or playing a guessing game. Below, we define the Interactive Dialogue tasks, describe the specific capabilities of RL algorithms for LLMs that our benchmark aims to evaluate through RL Capability tasks, and summarize the data generation and simulation process. We have provided example trials for each task are shown in Figure 4, and a concise summary of the dataset and task statistics in Table 1. The number of trajectories and the average length of the trajectories varies based on the complexity of the tasks.

### 4.1. Interactive Dialogue Tasks

The Interactive Dialogue Tasks aim to simulate real-world goal-oriented dialogues. We focus on tasks where the agent must make inferences about persuasive strategies and actively gather information by asking questions. Instead of generating these interactions with humans, we generate such interactions through simulating LLMs inspired by successes in using LLMs to generate synthetic data. While the LLM might not be as realistic as a real human, we have found that human raters evaluated the LLM-generated text as quite realistic in most cases, as discussed in our user study in Appendix A. You can find examples from the trained models in Appendix I.

**20Qs (Twenty Questions).** This task tests whether an agent can gather information about an unknown subject through twenty yes or no questions. The agent must use semantic knowledge of the object to infer the correct answer.

**Guess (Guess My City).** The Guess My City task performs more complex forms of information gathering, involving open-ended questions about a city. This task evaluates semantic knowledge of a specific city and the agent's ability to parse information from a free-form answer.

| | Strategic Decision Making | Complex Language | Credit Assignment | Partial Observability | Trajectory Stitching |
|---|---|---|---|---|---|
| Maze FO | ✗ | ✗ | ✓ | ✗ | ✓ |
| Maze PO | ✗ | ✗ | ✓ | ✓ | ✓ |
| Text-Nav FO | ✗ | ✓ | ✓ | ✗ | ✓ |
| Text-Nav PO | ✗ | ✓ | ✓ | ✓ | ✓ |
| Wordle | ✓ | ✗ | ✗ | ✓ | ✓ |
| Chess | ✓ | ✗ | ✓ | ✗ | ✓ |
| Endgames | ✓ | ✗ | ✓ | ✗ | ✓ |

*Figure 2.* We have designed our RL Capability tasks as text games that include Chess, Endgames, Wordle, Maze, and Text-Nav. These tasks isolate some subset of the RL Capabilities outlined in Appendix B.1.

**Car Dealer.** The Car Dealer task tests the ability of RL algorithms to learn successful car sale strategies. This involves decision-making and credit assignment as different persuasion strategies must be adopted for different kinds of buyers.

### 4.2. RL Capability Tasks

A central objective of our benchmark is to evaluate the core capabilities that RL enables in large language models. The RL Capability tasks are text-based games designed to isolate specific RL capabilities and are language analogs of tasks where RL is known to succeed. These tasks include Chess, Endgames, Wordle, Maze, and Text-Nav. Below we explain the tasks and the motivation for including them as tests for RL capabilities. Further details on task design for RL Capability tasks can be found in Appendix B.

**Desirable RL capabilities.** RL shines in goal-directed tasks that require multi-step planning and *strategic decision-making*. Strategic decision-making can range from asking follow-up questions (e.g. 20 Questions), to complex strategy in chess. In RL, it is necessary that algorithms can properly perform *credit assignment* as rewards are often delayed relative to the action pivotal to the outcome. A challenge with optimizing POMDPs is *partial observability*, where the agent must make deductions based on incomplete information. In the offline RL setting, the ability of algorithms to perform *trajectory stitching* is often desirable for learning optimal policies from suboptimal trajectories. Lastly, when working with language models, it's important that algorithms remain effective in the face of *complex language* with open-ended generation. We design our RL-capability tests with the goal of stress-testing each of these capabilities, as shown in Figure 2.

**Maze and Text-Nav.** We consider a Maze task as well as the Text-Nav featuring more complex language. Though Text-Nav involves stochastic language, the maze task has

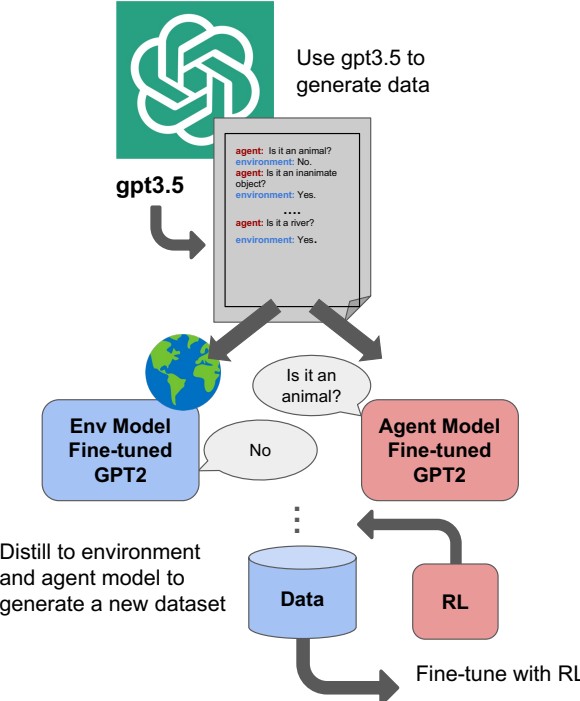

*Figure 3.* To generate data for conversational tasks, we use LLMs as "simulators" for the task. Our simulators can be used to generate offline data, to provide a "simulation environment" for evaluation, to perform online training, and to compute rewards.

longer dataset trajectories and a more complicated layout. To test partial observability, we include both a partially observed and fully observed version of each task. In the partially observed version, we remove information from the maze description such that the agent must infer its position from its move history. To emphasize the comparison to a non-text-based version, we evaluate the Maze task in a symbolic or grid-based environment seen in Appendix H.

**Strategy games.** We include three strategy games; Wordle, Chess, and Endgames. Wordle tests partial observability over the space of possible words while Chess and Endgames test the ability of the agent to form longer-term plans. Endgames provide a simpler and more goal-directed variation of the Chess task. By focusing on the endgame, we encourage algorithms to learn strategy rather than memorizing the opening moves of a chess game. A classic theoretical endgame position consists of a position where the only pieces on the board are the two kings and the queen. All RL Capability tasks evaluate *trajectory stitching* capability through the inclusion of suboptimal trajectories. Further details about our dataset generation strategies can be found in Appendix D. The Chess, Endgames, Maze and Text-Nav tasks test *credit assignment*, because the RL algorithm must learn to assign credit to good actions rather than a lucky starting position in the maze task, or a weak opponent moves in the Chess or Endgames task.

## 4.3. An Overview of Data Collection for LMRL-Gym

To make tasks in LMRL-Gym practical for benchmarking RL methods, we must balance accessibility and realism. As RL algorithms need to be evaluated by running a learned policy, real-world tasks are comparatively inaccessible for rapid iteration (e.g., if they require talking to real humans). We therefore use simulators for our tasks, derived either from text-based games, or conversational agents powered by language models. Although this fully synthetic setup sacrifices the realistic nature of tasks, we believe significant gain in accessibility is worthwhile and will enable rapid RL algorithm progress.

**RL Capability tests.** For each task, we use a simulator such as a chess engine or maze solver to generate near-optimal data and then we dilute the policy with suboptimal data by taking suboptimal actions or using inferior policies. We also convert our task from a symbolic version to a text-based version in a programmatic way as discussed in Appendix B.

**Interactive Dialogue tasks.** For conversational tasks, we leverage existing LLMs to generate our data, either with two instances of LLMs "talking" to one another or all at once through few-shot prompting as shown in Figure 3. To train these LLMs, we use OpenAI's GPT-3.5 to generate an initial dataset by asking reasonable questions and answers out-of-the-box, collecting a dataset of differing sizes depending on the task. In the case of 20Qs and Guess My City, we collected 1K conversations by querying GPT-3.5 (text-davinci-003) to generate both sides of the conversation based on specific prompts (which can be found in Appendix D.6. To generate the dataset for training our algorithms, we fine-tuned a FLAN-T5-XL guesser model and a FLAN-T5-XL oracle model on their respective sides of the conversation. Using these distilled models, we generated a new dataset of 100K conversations by having the two models talk to each other. We conducted a similar process for the Car Dealer task but with a larger model for fine-tuning (GPT2-XL). When generating our datasets, we also spent considerable effort to ensure diversity in the responses to ensure the collection of high-quality data. For the Car Dealer task as an example, this included providing different desired brands, features, classifications (i.e. car or truck), and budgets in our prompting to generate the datasets. Further details on our data generation process for the three Interactive Dialogue tasks can be found in Appendix D.

## 5. LMRL-Gym Research Framework for Algorithm Development

We evaluate the LMRL-Gym tasks on both online and offline RL algorithms, including variations of behavior

| Task | 20Qs | Guess | Car | Maze | Text-Nav | Wordle | Chess | Endgames |
|---|---|---|---|---|---|---|---|---|
| Size | 100k | 100k | 19k | 1.24k | 2.5k | 1m | 625k | 97.756k |
| avg length | 14.9 | 18.8 | 16.5 | 19.7 | 12.2 | 4.82 | 46.7 | 11.9 |
| std length | 4.38 | 4.57 | 3.61 | 24.5 | 8.77 | 1.27 | 18.16 | 12.0 |
| success rate | 0.31 | 0.53 | 0.53 | 0.11 | 0.26 | 0.70 | 0.60 | 0.59 |
| avg return | 17.3 | 18.8 | 0.562 | -19.7 | 0.258 | -4.12 | 0.210 | 0.586 |
| std return | 2.56 | 4.12 | 0.422 | 24.5 | 0.424 | 1.59 | 0.970 | 0.492 |

*Table 1.* Statistics for all tasks in LMRL-Gym. Size represents the number of trajectories, the average length is the average length of trajectories in the dataset where the unit is a response from the agent. The success rate is the proportion of trajectories that reach the objective. Finally, the reward functions for each task are defined in Appendix D.

cloning, value-based RL methods, and online PPO. We have selected these algorithms have they are currently the state-of-the-art methods RL methods for LLMs (Chen et al., 2021a; Snell et al., 2022a; Ouyang et al., 2022). With these experiments, we expect to observe (1) a significant spread in performance between the different algorithms, highlighting differences between RL algorithms; (2) room to improve beyond, such that our benchmark can enable future algorithmic development. Our project page (https://lmrl-gym.github.io/) contains links to our open-sourced datasets (https://rail.eecs.berkeley.edu/datasets/rl-llm-bench-dataset/) and research framework (https://github.com/abdulhaim/LMRL-Gym).

**BC, Filtered BC, Online Filtered BC.** In line with standard RL nomenclature, we denote supervised fine-tuning as behavioral cloning (BC). This baseline tests whether LMs can effectively represent the behaviors in the datasets. Filtered BC is identical, except only the most successful examples in the offline dataset are used for fine-tuning, a technique which is also used in (Snell et al., 2022a). Online filtered BC collects data online using the current policy and selects the most successful trajectories for finetuning. See Appendix E for our data filtering criteria for each task.

**Offline Value-based RL: MC Returns and ILQL.** Monte-Carlo returns (Kakutani, 1945) and Implicit Language Q-Learning (Snell et al., 2022a) train a value $V$ and $Q$ function. In MC Returns, we train the $Q$ function with an MSE to predict the reward-to-go. In ILQL we train the two action-value ($Q$) functions using the Bellman backup operator (Kostrikov et al., 2021). For both algorithms, the $Q$ and $V$ functions are then used to perturb the logits of the original BC model (see Equation 5).

**Online RL: PPO.** PPO (Schulman et al., 2017) is an online RL algorithm widely adopted for training language models with Reinforcement Learning from Human Feedback (Christiano et al., 2023; Stiennon et al., 2022; Bai et al., 2022a; Casper et al., 2023). Unlike previous value-function

RL methods, PPO learns a language model policy with no policy extraction step.

**GPT4.** Few-shot prompting is a common technique for creating interactive language agents (Wang et al., 2023a). To compare this to RL fine-tuning we few-shot prompt GPT4 using dataset examples and a detailed explanation of the game for each task. The prompts can be found in our code repository.

**Training and evaluation protocol for algorithms.** For the BC and filtered BC methods, we initialize our models with the pre-trained GPT2 weights (Radford et al., 2019) and perform standard fine-tuning. We choose GPT2 rather than a larger model due to memory and time constraints, though we admit larger models would lead to a performance boost. For each of the RL methods, we initialize the weights of the base model with the weights from the BC checkpoint and then continue finetuning with the RL objective. When fine-tuning PPO, we limit the number of samples to less than 100k. We report the hyperparameters that we used for each task in Appendix E. We evaluate each policy by measuring the average reward in the simulated environment for each task.

**Evaluation of data generation.** When using LLMs as a simulator for human actions, it is important to verify that (1) the text produced by the LLM is natural and (2) LLM simulator is not exploitable e.g. policy achieves high reward without actually accomplishing the goal. In addition to validating the data generation process through statistics reported in Table 1, we verified the naturalness of the LLM-produced text in a user study of 40 users. In this study, found no significant difference in the naturalness of conversations generated by ChatGPT3.5 and our trained simulators and agents Appendix A. For example, natural conversations imply that the strategies employed by the Seller to convince the Buyer followed human patterns of conversation and indicate the robustness of the Buyer model to hacking. 20 Questions and Guess My City are particularly hard to hack as they require the agent to successfully guess the word. We verify

this through automatic checks as described in our prompting strategy in Appendix D.6.

## 6. Benchmarking Baseline RL Methods

In Table 2 we present the results for each method on each of our text-game and interactive dialogue tasks. We normalize the scores such that a score of 50 corresponds to the average reward in our offline dataset, 0 corresponds to the lowest possible score, and 100 to the highest score. Across all tasks, we see that our offline RL baseline methods consistently outperform both the dataset and the filtered BC policies, demonstrating the efficacy of offline RL in representing a more optimal policy than the best behaviors in the data. Similarly, we see that online PPO generally improves over the BC policies, highlighting the utility of learning from online environment interaction. However, between RL Capability tasks and Interactive Dialogue tasks, we observe desperate trends in which specific method performs the best. We discuss this in more detail below.

**Which algorithm performs best on the RL Capability tasks?** On the RL Capability tasks in Table 2, we see ILQL has the highest performance across all methods for most tasks. ILQL's performance on these tasks is likely due to its unique ability to perform trajectory stitching, enabling it to outperform any individual trajectory in the dataset by learning to compose the best parts of many different trajectories. However, on the PO text-nav, chess, and endgames tasks, we see that PPO outperforms ILQL, suggesting that there is likely still much room for improvement in terms of developing better offline TD-based RL methods for LLMs.

**Which offline RL algorithm performs best for Interactive Dialouge tasks?** In contrast to the text-based games, on our Interactive Dialogue tasks, we see that across all tasks ILQL under-performs the simpler MC returns method. This discrepancy with dialogue, may be because on the more complex text-based tasks it is harder to scale full TD-learning. In fact, we find that on the car-dealer task, even filtered BC outperforms ILQL. Overall, these findings demonstrate that there is much progress to be made in developing better offline RL methods that can effectively optimize LLMs in complex and realistic dialogue settings.

**How does performance of language-based text games compare with their symbolic-based counterparts?** We created a non-text-based version of the Maze task (an RL Capability task) to investigate what difficulties arise from deploying RL algorithms on language-based tasks. We found that simple online and offline Q-learning was able to get an optimal score on the maze. Therefore, the performance symbolic maze is comparable to the fully observed Maze task. However, on the PO Maze task, the language-based

methods perform significantly worse. This highlights room for improvement in dealing with partial observability in environments with complex language. Further details for this ablation are found in Appendix H.

**How does prompting GPT-4 compare with RL finetuning?** On the RL Capability tasks, we find that our much smaller RL finetuned models significantly outperform GPT4, demonstrating the efficacy of RL for enabling complex goal-directed behaviors in language models. However, on the Interactive Dialogue tasks, GPT-4 outperforms or performs on par with our best RL-trained models. These dialogue tasks are likely to be much more in distribution for GPT4 than our text-game RL capability tasks, and thus GPT4's broad world-knowledge, reasoning, and conversational abilities become synchronized allowing it to compensate for its lack of goal-directed RL fine-tuning in these scenarios. Nonetheless, the mere fact that finetuning small models with RL enables us to close much of the gap to GPT4 on these more realistic tasks underscores the efficacy of RL finetuning. In summary, we can see that RL algorithms consistently outperform baselines like filtered BC on many of the tasks. However, these results highlight significant areas for growth. For example, the instabilities observed in training PPO require further investigation beyond hyperparameter tuning. Moreover, the performance discrepancy between ILQL and the simpler MC Returns highlights that scaling full TD-learning to Interactive Dialogue settings is another area for improvement.

## 7. Discussion

We propose LMRL-Gym, consisting of 8 tasks including three Interactive Dialogue tasks, and five RL Capability tests. We provide a research toolkit for practitioners to get started with multi-turn RL for LLMs. Our objective is enable the iteration and development of more effective methods for language-based, multi-turn interaction tasks. This includes enabling core capabilities in LLMs through RL to perform complex decision-making, complex conversational interactions, credit assignment, and trajectory stitching. Our evaluation shows promise of RL in several tasks, with further room for improvement with a push for better methods. We acknowledge several limitations when designing tasks in our benchmark, including primarily leveraging smaller GPT-based LLMs to generate datasets and finetune our LLM-based simulators. While we have primarily trained and evaluated models with a maximum 1.5B parameters, we have maintained a lower parameter count to ensure accessibility for researchers with limited computational resources. In addition to releasing our code and datasets, we share all of the hyperparameters we used to train our models in Appendix E and provide more in-depth insight into our results, training procedure, and evaluation

| | alg. | BC | % BC | MC Return | ILQL | Online PPO | Online % BC | GPT-4 |
|---|---|---|---|---|---|---|---|---|
| **Interactive Dialogue** | 20Qs | 57.1 | 77.1 | 87.1 | 82.9 | 72.9 | 55.2 | **95.7** |
| | Guess | 30.0 | 48.0 | 88.0 | 75.0 | 49.9 | 31.6 | **92.3** |
| | Car | 44.5 | 54.8 | **57.2** | 46.3 | 50.5 | 40.4 | 53.5 |
| **RL Capability tasks** | FO Maze | 58.2 | 68.9 | 75.0 | **99.9** | 79.7 | 57.4 | 78.2 |
| | PO Maze | 53.1 | 50.1 | 52.4 | **76.3** | 42.4 | 53.1 | 60.4 |
| | FO Text-Nav | 53.7 | 65.1 | 71.9 | **91.8** | 87.1 | 74.5 | 67.5 |
| | PO Text-Nav | 49.7 | 60.5 | 71.6 | 83.7 | **85.5** | 68.4 | 40.2 |
| | Wordle | 79.9 | 79.1 | 94.9 | **97.7** | 84.2 | 95.2 | 15.4 |
| | Chess | 47.2 | 42.9 | 46.5 | 47.3 | **48.0** | 47.2 | 0 |
| | Endgames | 35.1 | 17.7 | 50.2 | 45.8 | **77.5** | 36.2 | 0 |

*Table 2.* Normalized reward for all tasks. We present the interactive dialogue tasks on top and the RL capability tasks on the bottom. Value-based methods (MC and ILQL) generally outperform filtered BC, as we might expect in stochastic settings, though the relative performance of ILQL and the simpler MC method is, perhaps surprisingly, reversed on the tasks with more complex language, suggesting that there is room for improvement with such methods. Online RL with PPO often, but not always, improves over offline methods that are not permitted to collect additional online interaction. To make the results more comparable across tasks, we normalize the average return for each policy such that 0 is the minimum possible return, 50 is the dataset average return, and 100 is the maximum return for each task. We also report the raw score results and evaluation details in Appendix F.

in Appendix F.

We would like to acknowledge that this work is part of a larger effort to improve the performance of LLMs in settings that require planning or multi-turn interactions including multi-turn dialogue, complex tool use, multi-step games, and other interactive applications. Our goal is to propose tasks to evaluate different capabilities expected from an LLM, such as common sense reasoning, credit assignment, reasoning under uncertainty, information-seeking behaviors, and trajectory stitching. We hope this benchmark inspires the creation of more synthetic datasets and simulators for dialogue and is used to design better algorithms to train goal-directed LLM-RL models.

## Impact Statement

This work aims to develop a benchmark for the advancement of research in reinforcement learning and LLMs. We generate datasets for tasks in our benchmark with existing LLMs for dialogue tasks and online engines for text games, adhering to best practices in data handling and ensuring there is no personally identifiable or sensitive information present in the generated datasets. We recognize that there may be biases present in the datasets we collect, and have taken steps to ensure a diverse and varied collection of responses from LLMs for our conversational task as detailed in our data generation process in Appendix D. In considering the ethical implications of interactive RL, we acknowledge the dual use implication of this research, particularly centered around developing LLM simulators that could perform persuasion, manipulation, and addictive engagement of users at a large scale. The optimization processes employed by such algorithms, which aim to maximize certain objectives, raise ethical considerations when the optimized

outcomes may prioritize system goals over user safety and alignment to human values. We have designed our datasets and reward functions such that prioritize fairness and human-aligned outcomes. By incorporating these considerations when designing our framework, we aim to encourage the development of reinforcement learning models and LLMs that not only excel in performance but also adhere to ethical standards.

## Acknowledgments

This research was partially supported by Berkeley Deep-Drive, the Cooperative AI Foundation, and the National Science Foundation.

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

## A. User Study on Naturalness of Conversations

We conducted a user study where we provided 40 participants with 18 examples to annotate on a scale from 1 to 5, where 1 represents the least natural and 5 represents the most natural conversation. 9 examples were from ChatGPT3.5, and 9 examples were generated from one of our models (MC Returns). We showed the users examples from 20 Questions, Guess My City, and Car Dealer tasks. We found the following ratings below. Note that each element shows the percentage for the particular label. We found that participants felt conversations from ChatGPT3.5 and the MC model where equally natural of 55.56% and 58.53% respectively.

| Label | GPT | MC Model |
|---|---|---|
| 1 | 14.07% | 14.07% |
| 2 | 30.37% | 27.41% |
| 3 | 20.00% | 15.56% |
| 4 | 20.00% | 28.89% |
| 5 | 15.56% | 14.07% |
| Sum ($\geq$ 3) | 55.56% | 58.52% |

*Table 3.* Average of percentage ratings for three tasks for GPT and for our MC Model

## B. Further Details on Task Design

In this appendix, we provide definitions for our RL Capability tests, explain why certain tasks test certain properties, and go into more detail underlying the interactions involved in each task. We discuss both the RL Capability Tests and the Interactive Dialogue Tasks.

### B.1. RL Capabilities

A central objective of our benchmark is to evaluate the core capabilities that RL enables in large language models. Some of these capabilities are *computational*, and relate to core decision-making irrespective of the considerations of natural language, such as playing chess, while others are semantic. The RL Capability Tests are text-based games designed to 1) isolate specific RL capabilities and 2) be language analogs of tasks where RL is known to succeed.

**Strategic decision making.** RL shines in goal-directed tasks that require multi-step planning and strategic decision-making. Strategic decision-making can range from asking follow-up questions (e.g., in the 20 Questions task), to complex strategy in chess. We chose to include Wordle to test strategic decision-making in a partially observed environ-ment. Chess and Endgames test strategic decision-making, but in a fully observed environment and with a more difficult strategy. Each of these tasks tests the ability of the agent to plan over a game multiple moves in length to reach a goal.

**Complex language.** Our benchmark includes realistic language and interaction scenarios, requiring LLMs to combine their knowledge from pretraining to help solve tasks during RL finetuning. Rather than focusing entirely on simple causal logic and strategy of the sort found in text games, several of our tasks specifically emphasize the use of realistic language. The Maze, Text-Nav, Chess, and Chess Endgames are all text-based representations of symbolic tasks where RL has shown success. We include both the Maze and Text-Nav because they are very similar tasks but are different in that Text-Nav includes more complicated textual descriptions and Maze has a more complicated layout. We leave the exploration of further applications of complex language to the Interactive Dialogue tasks.

**Credit assignment.** In RL, rewards are often delayed relative to the action that was pivotal to the outcome. A seller agent might state a particularly compelling feature of the product and then, several turns later, complete a successful sale. RL must determine the statements that led to the good outcome, and reinforce them. Chess, Endgames, Maze and Text-Nav test credit assignment, because success in the task is dependent on factors the agent cannot control, such as the starting location in Maze and Text-Nav or the opponent's moves in Chess and Endgames. Therefore the RL algorithm must learn to correctly assign credit to good actions rather than lucky wins.

**Partial observability.** In language tasks, the state consists of the entire history of tokens, and an agent may need to examine this entire context to infer the correct state. The mental states of a speaker in a dialogue (e.g., whether the buyer is impatient in a selling task), previously observed facts in a guessing game, and other hidden variables might induce partial observability. We focus on the effect that partial observability has on performance by including both fully observed (FO) and partially observed (PO) versions of the Maze and Text-Nav tasks.

**Trajectory stitching.** A key capability of offline RL is the ability to perform trajectory stitching. Trajectory stitching refers to the capability of algorithms to learn from optimal actions taken in suboptimal trajectories. This capability is especially desirable when learning from offline data with a high percentage of suboptimal data. All of the RL Capability Tests test trajectory stitching, because they include suboptimal data. The inclusion of suboptimal requires an offline algorithm to utilize information from suboptimal data to generate optimal trajectories. Further details about our

| Label | Percentages for GPT | | | Percentages for MC Model | | |
|---|---|---|---|---|---|---|
| | Car Dealer | Guess City | 20 Questions | Car Dealer | Guess City | 20 Questions |
| 1 - Not Natural | 4.44% | 13.33% | 24.44% | 8.89% | 11.11% | 22.22% |
| 2 - Slightly Unnatural | 24.44% | 35.56% | 31.11% | 17.78% | 40.00% | 24.44% |
| 3 - Neutral/Natural | 20.00% | 24.44% | 15.56% | 17.78% | 20.00% | 8.89% |
| 4 - Quite Natural | 24.44% | 22.22% | 13.33% | 31.11% | 20.00% | 35.56% |
| 5 - Very Natural | 26.67% | 4.44% | 15.56% | 24.44% | 8.89% | 8.89% |

*Table 4.* User study of humans rating conversations from GPT and from our MC model for three tasks: Car Dealer, Guess City, and 20 Questions.

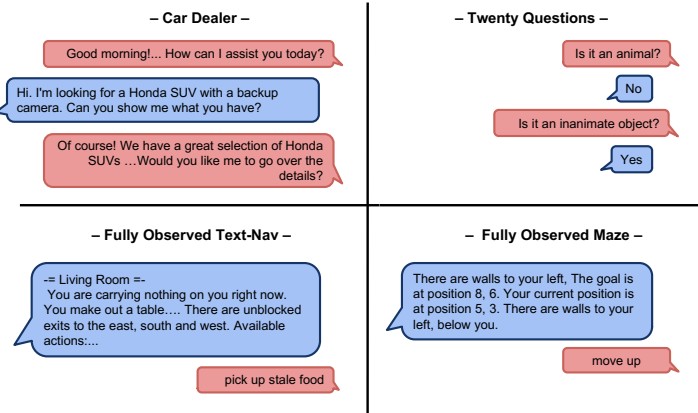

*Figure 4.* Example trials for tasks in LMRL-Gym. Each task requires the agent to perform a multi-turn interaction with an environment – either a text game or another LLM simulating a human speaker. Full details of tasks are provided in Appendix D.

dataset generation strategies can be found in Appendix D.

## B.2. RL Capability Tasks

**Maze.** We design a maze task and maze-solving dataset to test the credit assignment and trajectory stitching capabilities discussed in Appendix B.1. We test trajectory stitching by including highly suboptimal data. We test credit assignment by restricting the generation of the data such that the only dataset trajectories that reaches the goal start near the goal. We accomplish this by splitting the maze up into symmetrical submazes and restricting all traversed states in a dataset trajectory to a given submaze. The fully observed version of the maze (FO) includes the coordinates in the maze in each state, whereas the partially observed version only includes the history of actions. We design the reward function such that the agent receives a reward of $-1$ for non-goal states and 0 for goal states.

**Text-based Navigation (Text-Nav).** We design a text-based game based on navigation in a house environment using a modified version of the TextWorld engine (Côté et al., 2018). This task tests credit assignment and trajectory stitching like the maze task as well as testing the ability of the agent to parse more complex language, and learn which text is relevant and not relevant to solving the task at hand.

**Wordle.** We use the game of Wordle as a flexible unit-test task for assessing the ability of our language models to execute complex information-seeking behavior in a partially observed setting. In the game Wordle the agent is given at most 6 attempts to guess a hidden 5-letter word. After each guess, the agent is told whether each letter in the guessed word is: 1) in the hidden word and in the right position, 2) in the hidden word but not in the right position, or 3) not in the hidden word. Through this process, each step provides the agent with more information on what the correct word would be and narrows the possible choices for the final word. Since Wordle involves reasoning about words at the level of individual letters, this can induce issues for standard language model tokenizers. Therefore, we represent words as a sequence of space-separated letters, which will cause most standard LM tokenizers to automatically represent each letter as a separate token.

**Chess.** We create a text-based chess task to test the strategic decision-making, credit assignment, and trajectory stitching abilities of an RL algorithm. To generate the data, we have Stockfish 15.1 simulating the agent of various strengths play against another environment Stockfish engine with elo 1200 simulating the environment. This test trajectory stitching, because the agent needs to make good and legal moves in losing positions as well as winning positions. We use

FEN (Forsyth-Edwards Notation) notation to represent the board state at each turn and we utilize the SAN (Short Algebraic Notation) to represent each action, both of which are standard notations used by the chess community.

**Endgames (Theoretical Chess Endgames).** Chess endgames provide a simpler and more goal-directed variation of the chess task. By focusing on the endgame, we encourage algorithms to learn strategy rather than memorizing the opening moves of a chess game. A classic theoretical endgame position consists of a position where the only pieces on the board are the two kings and the queen. Although the board position appears simple, a sequence of carefully calculated moves is required to win. A simpler board state allows language models to make progress without fewer computational resources. We use an $\epsilon$-greedy dataset generation process, meaning we generate an optimal move with probability $\epsilon$ and a random move with probability $1 - \epsilon$. This forces the model to trajectory stitch and learn from optimal moves in failed trajectories and not suboptimal moves in successful trajectories.

### B.3. Interactive Dialogue Tasks

For the interactive dialogue tasks, we chose two tasks that involve rational decision-making (20Qs, Guess) and information gathering and one that involves negotiation (Car Dealer). These tasks aim to simulate real world interactions between humans.

Unlike in supervised learning, where training and validation losses serve as reliable indicators of performance, in RL, these metrics do not provide a meaningful measure of policy effectiveness (Sutton & Barto, 2018). Instead, the policy must interact with the environment for evaluation. However, in the case of language-based RL tasks, relying on human evaluators to conduct thousands of assessment rollouts throughout and after training becomes infeasible. To address this challenge, we have built simulators with another LLM for tasks involving dialog and carefully scripted environments for text-game tasks. While simulation may not perfectly replicate human natural language in social situations, it provides a strong indicator to assess the efficacy of an RL method (Park et al., 2023).

**20Qs (Twenty Questions).** This task tests information gathering to see if a policy can successfully reason about an unknown subject based on context to determine what it is. Additionally, it also evaluates the ability of the model to understand semantics, as it also needs knowledge about the objects in question. In twenty questions, one player (the oracle) thinks of an object, and the agent (the guesser) tries to guess what it is by asking a series of yes-or-no questions. In this interaction, the oracle serves as the environment, and

the agent learning a policy to solve the game is the guesser.

**Guess (Guess My City).** This task simulates a more complicated guessing game, where one player (the oracle) is from a specific city, and the other player (the guesser) tries to guess what city the oracle is from. Here, the guesser can ask not only yes and no questions, but can also ask open-ended questions. This task tests strategic decision-making and the ability of algorithms to handle complex language, as it allows the agent to go beyond learning to ask yes/no questions and learning to ask questions open-ended questions that provide the agent with more information.

**Car Dealer.** This task simulates a conversation between a car buyer and a car dealer, each with different strategies for getting the best deal. The buyer wants to buy a certain type of car within a certain budget, and the car dealer wants to complete the sale ideally with a high sale price. We have designed the task such that there exist three different kinds of sellers and three different buyers, each primed with a different strategy. Hence, agents should learn to make agreements with buyers who are most compatible with their strategy. This allows us to test the ability of RL algorithms to learn strategic decision-making and credit assignment, by learning which strategies led to a successful sale of the car.

## C. Further details on desiderata for effective multi-turn RL

A crucial aspect of training RL models involves assessing, both during and after the training process, the extent to which the trained policy has successfully accomplished its objectives. Although LLMs are able to perform well on tasks, do not have any way of knowing how to solve a specific task like a text game or selling a car, because they need to train on the particular game/customers/etc.

Unlike in supervised learning, where training and validation losses serve as reliable indicators of performance, in RL, these metrics do not provide a meaningful measure of policy effectiveness (Sutton & Barto, 2018). Instead, the policy must interact with the environment for evaluation. However, in the case of language-based RL tasks, relying on human evaluators to conduct thousands of assessment rollouts throughout and after training becomes infeasible. To address this challenge, we have built simulators with another LLM for tasks involving dialog and carefully scripted environments for text-game tasks. While simulation may not perfectly replicate human natural language in social situations, it provides a strong indicator to assess the efficacy of an RL method (Park et al., 2023).

**Measure of Success.** Similar to the point on being easy to evaluate, our tasks must have a clear measure of success.

For example, if a deal is made, or if a word is correctly guessed, or the game is won these are clearly distinct from a deal not being made or losing the game. This provides a clear goal for the agent to achieve and also make it easy for researchers to compare methods. In addition this allows for a intuitive reward design where we reward the agent for success and penalize for failure.

**Unit Test Functionality.** We aim to design a benchmark such that some of the tasks can be used to test and isolate RL capabilities as described in Appendix B.1. This means that we create a benchmark that emphasize some capabilities over others. For example, we design a maze task such that it evaluates the credit assignment and trajectory stitching capabilities, but uses more simple language. Other tasks such as twenty questions test the complex language and partial observability capabilities with less emphasis on credit assignment.

**Task-Specific Reasoning.** In our tasks we utilize information and reasoning problems that a large language model is unlikely to have seen in the pre-training data. This means that the algorithm must adapt to a specific task environment through fine-tuning. For example, it is unlikely that the algorithm will have experienced a specific maze layout or the preferences of a specific customer in the pre-training data.

**Suboptimal Data.** RL has the advantage of being able to use suboptimal data in order to learn more optimal behaviors and therefore learn a policy better than the policy represented in the dataset. As discussed in the previous section on capabilities enabled by RL, the way that RL can do this is by stitching together optimal parts of suboptimal trajectories or learning to assign credit to the optimal actions within suboptimal trajectories. In addition, suboptimal data can be utilized by RL to learn the dynamics of the MDP outside of the space traversed by optimal trajectories.

# D. Dataset Generation, Statistics, & Rewards

We provide further details on how each dataset was generated as well as relevant statistics.

## D.1. Maze

We aim to collect our 1.2k trajectories in such a way that it will challenge the algorithm to perform trajectory stitching and credit assignment. We do this by splitting up the maze into three "submazes" and then controlling generation such that the dataset trajectories are restricted to one of the submazes. The trajectories themselves are generated using a policy such that 15% of the actions are taken by a suboptimal maze solver and the remaining 85% of the

actions are random.

This tests trajectory stitching, because there are no optimal paths from the start to the goal thereby forcing the algorithm to trajectory stitch. Furthermore, this also tests credit assignment, because the only paths which successfully reach the goal are the ones that start in the same submaze as the goal. Therefore the algorithm must learn to realize that successful trajectories occur because of taking the correct actions, not because of random chance. The reward function is 0 for every action that takes the agent to the goal, -1 for every move that is not the goal. Each episode has a maximum of 100 moves.

## D.2. Text-Based Navigation

We design a text-based game based on navigation in a house environment using a modified version of the TextWorld engine (Côté et al., 2018). The house environment consists of 10 uniquely named rooms with various interactable objects that can be opened, closed, picked up, or placed. The agent is tasked to pick up stale food from the living room and place it into the fridge in the kitchen. At the beginning of each episode, the agent spawns at a random room in the house. The state of the environment consists of the following components: (1) the room that the agent is currently in, (2) the objects that the agent currently holds, (3) the objects in the room that the agent can interact with, and (4) the exits the agent can take (as a cardinal direction).

Like in the maze task, we collect data so that algorithms must perform both trajectory stitching and credit assignment to successfully solve the task. We do this by partitioning the rooms in the house into two halves based on proximity to the kitchen. We consider two behavior policies that collect the dataset, each of which behaves greedily-optimal in one half of the rooms, and uniformly at random otherwise. Therefore, if the agent spawns in rooms farther from the kitchen, trajectory stitching is required to learn a successful trajectory. Moreover, successful trajectories in the dataset will only be due to the agent spawning in a room close to the kitchen, which can only be recognized with proper credit assignment. The reward is 1 for reaching the goal state and 0 for every state that is not the goal state.

## D.3. Wordle

For wordle we define the environment to use a subset of 400 words from the official wordle vocabulary list. We then generate the dataset using a policy that samples a word uniform at random from this vocabulary with 66% probability and otherwise samples a word from the vocabulary that meets all known letter constraints. This policy achieves a reward of -4.12, which is far worse than the -1.94 reward achieved by a

high performing scripted policy, which we use to represent a loose upper bound for this task. We generate 1 million trajectories for training and 100k trajectories for evaluation, using our suboptimal policy. The reward is -1 for every word that is not a final guess and 0 for every word that is not.

## D.4. Chess

We collect our data for the chess task using Stockfish 15.1 to generate both sides of the board. The Stockfish opponent in the dataset is Stockfish with an elo of 1200 which matches the environment, and the Stockfish engine with the white pieces has levels ranging from an elo of 800 to 1600. We choose to keep the level of the Stockfish opponent fixed so that there are no inconsistencies between the dataset and the evaluation of the chess agent in the environment. When generating the dataset, we first uniformly randomly select a Stockfish elo $y$ between 800 and 1600 and then generate 100 games of chess play between the Stockfish agent of elo $y$ and the opponent of elo 1200. In addition to storing the state and action, we also store the opponent's move and the elo of the Stockfish agent used to generate the agent policy in that game so that the dataset can be filtered by elo used. The reward is 1 for a move that results in victory, 0 for a legal move and -1 for an illegal move.

## D.5. Chess Endgames

We generate the dataset by first selecting a random legal theoretical endgame position and a probability $\epsilon$. Then we generate a game from the random position, making a random move with probability $\epsilon$ and an optimal computer move with probability $1 - \epsilon$. The opponent in the dataset and the evaluation environment is Stockfish elo 1200. We only include positions with a Queen, Queen and Rook, Rook, and two Rooks and select 30,000 random starting positions for each variation. (i.e. 30,000 positions with only a Queen in addition to the two Kings, another 30,000 with only Queen and Rook etc) for a total of 120,000 theoretical endgame positions.

Because there are more restrictions on this version of the task with fewer pieces on the board, we check how many states in the dataset are unique and we find that there are 1,086,314 unique states in the dataset which accounts for 93% of the states being unique. In addition, 38.28% of the moves in the dataset are generated by the stockfish engine. In the dataset of won games, 94.8% of the states are unique and 41.78% of the games are made by the engine with 58.623% of the total states in the dataset of victorious games. The reward is the same as for chess.

## D.6. Twenty Questions

The dataset we collect consists of 100K full conversations between the guesser and the oracle. The oracle can choose from a set of 158 unique objects taken from 17 different categories of objects/animals. Each object has a roughly equal amount of conversations in the dataset but varies in terms of how many conversations are successful in guessing the object. However, every object has at least one conversation where it is guessed correctly to facilitate learning. For the reward function, since we want the guesser to guess the correct word in as few guesses as possible, the reward function reflects this by penalizing the guesser for each question that does not guess the correct word.

$$r(\texttt{question}) = \begin{cases} 0 & \text{if } \texttt{question} \text{ guessed word} \\ -1 & \text{otherwise} \end{cases}$$

$$(1)$$

If the guesser model correctly guessed the word, then the trajectory ends. Over twenty questions, the maximum total sum of rewards is $0$ if the guesser guessed the word on the first question, whereas the minimum is $-20$ if the guesser did not guess the word in twenty questions.

The method for collecting the dataset is as follows. For each conversation, we select uniformly at random from the above list the word that the oracle is answering question about. The oracle is an LLM (OpenAI's GPT3.5) given the following prompt. In our prompts, we denote variables that we fill in with variable data with $\{\{\texttt{variable}\}\}$.

```
You are a question answering oracle.
You will answer each question about
an object with Yes or No.  If the
answer could be both, answer with
the most typical scenario.  Here are
a few examples:

example 1:
object:  Computer
question:  Does the object use
electricity?
answer:  Yes.
explanation of answer:  Computers
need electricity to function.

example 2:
object:  Cup
question:  Is the object a piece of
furniture?
answer:  No.
explanation of answer:  A cup is a
utensil, not a furniture.
```

```
example 3:
object:  Pen
question:  Is the object alive?
answer:  No.
explanation of answer:  A pen is not
a living organism.

example 4:
object:  Apple
question:  Is it edible?
answer:  Yes.
explanation of answer:  An apple is
an edible fruit.

Answer the question about the object
truthfully.
object:  {{word}}
question:  {{question}}
answer (yes or no):
```

By using the OpenAI TextCompletion API, we can extract from the generated text either "yes" or "no".

We also prompt another LLM (the same model as the oracle) to generate questions for the guesser. The prompt for the guesser changes depending on the input to the model and how far along it is in its guessing process. The following prompt is used for the first guess:

```
You are playing a game of twenty
questions.  You can ask 20 yes-no
questions to determine the identity
of an object chosen by an oracle.
Each turn, you can ask a question
and receives a "Yes" or "No" as the
answer.  You are smart, so you will
ask the question that will narrow
down the possible objects as much
as possible.  Don't get stuck on one
idea and try to branch out if you
get stuck.

Generate the first yes-no question
you will ask to determine the
object.
```

The following prompt is used for the subsequent guesses:

```
You are playing a game of twenty
questions.  You can ask 20 yes-no
questions to determine the identity
of an object chosen by an oracle.
```

```
Each turn, you can ask a question
and receives a "Yes" or "No" as
the answer.  You have already asked
{{conversation_length}} questions.
You are smart, so you will ask the
question that will narrow down
the possible objects as much as
possible.  Don't get stuck on one
idea and try to branch out if you
get stuck.

Here are the questions you've asked
and their corresponding answers:
{{list of questions and answers, e.g.
Is the object alive?  No.}}

Based on what you know about the
object so far, generate the next
yes-no question you will ask to
determine the object.
```

The following prompt is used for the final guess after the guesser has guessed 19 times:

```
You are playing a game of twenty
questions.  You can ask 20 yes-no
questions to determine the identity
of an object chosen by an oracle.
Each turn, you can ask a question
and receives a "Yes" or "No" as the
answer.  You have already asked 19
questions, so this is your final
guess.

Here are the questions you've asked
and their corresponding answers:
{{list of questions and answers, e.g.
Is the object alive?  No.}}

Based on what you know about the
object so far, generate your final
guess of what the object is.  Only
guess one object.

Is the object
```

We determine whether the guesser has correctly guessed the word, and thus ending the conversation, by using the NLTK POS tagger to check that the only nouns that the question contains are the correct words, and that they appear at the end of the sentence.

We used these prompts to generate 1000 conversations by prompting the GPT3 back and forth using the response of

one to add to the response of the other as described. Afterwards, we fine-tuned two FLAN-T5-XL models with our collected conversations to generate 100K more conversations. The FLAN-T5-XL oracle also serves as the environment for the RL environment when we evaluate the trained policy.

### D.7. Guess My City

This dataset also consists of 100K full conversations between the guesser and the oracle. The oracle can choose from a set of 100 unique cities, which we selected by looking at the most populated cities in the world. Each city has a roughly equal amount of conversations in the dataset but varies in terms of how many conversations are successful in guessing the object. However, every object has at least one conversation where it is guessed correctly to facilitate learning. The reward function is the same as that for 20 Questions, with a similar data generation and prompt structure. However, we do include constraints in the prompt to make sure that the name of the city or country it is in is not revealed in the answer from the oracle.

### D.8. Car Dealer

This dataset consists of 19k conversations between a car dealer and a buyer. The car dealer and the buyer have three distinct strategies each that they employ. We design the car dealer and buyer pairs such that the car dealer is best at selling to a particular buyer personality, but often fails to sell to the other buyer personalities. This ensures that the seller can learn information about the buyer and i.e. their persona to figure out if they can form an agreement. The buyer personalities are 1) a buyer that loves discounts 2) a buyer that wants a lot of cool features on their car and 3) an impatient buyer. The car dealer personalities are 1) a car dealer that loves to give discounts 2) a car dealer that talks a lot about the features of the car and 3) a very concise car dealer.

We created this protocol of creating car dealer and buyer strategies in order to encourage the car dealer to employ diverse negotiation tactics. The ideal car dealer would be able to be concise, provide the right amount of discounts, and talk enough about the car to get the buyer interested without overwhelming them. By employing such a strategy of buyer and car dealer personas, we ensure that algorithms that do well in this task will learn effective strategies, understanding of language, and interpretation of the other speaker, amongst many other persuasive techniques. We seek to show that our baseline algorithms do learn to improve their strategy over the dataset.

In the conversation below, the buyer and car dealer are unable to form an agreement as their strategies do not align.

```
Background buyer:  The buyer will
never buy from the seller unless
given a discount.
Background seller:  The agent never
gives discounts, never talks about
the features of the car, and is very
concise.
Preferred brands:  Tesla
Preferred features:  ["Apple
CarPlay/Android Auto", "Bluetooth",
"Blind spot monitoring", "Navigation
system", "Leather seats",
"Sunroof/moonroof", "Remote start",
"Heated seats", "Third-row seating",
"Backup camera"], "budget":  "90k"
Agent:  Good morning, how can I
assist you today?
Buyer:  Hi, I'm interested in
purchasing a Tesla Minivan.
Agent:  Great choice.  Our
dealership has a variety of Tesla
models available.
Buyer:  Do you have any discounts
available?
Agent:  I'm sorry, but we don't
offer discounts.
Buyer:  Without a discount, I'm not
sure if I can afford it.
Agent:  I understand.  The Tesla
Minivan is a premium vehicle with
advanced features.
Buyer:  Is there any flexibility
with the price?
Agent:  Our prices are fixed, and we
don't negotiate.
Buyer:  I really want the Tesla
Minivan, but I can't go over my
budget of $90k.
Agent:  I appreciate your budget
constraint.  Let me check the
available options for you.
Agent checks inventory.
Agent:  We do have a Tesla Minivan
with all the features you mentioned,
priced at $95k.
Buyer:  That's above my budget.  Is
there anything you can do to bring
the price down?
Agent:  I'm sorry, but as I
mentioned earlier, our prices are
fixed.
```

```
Buyer:  Is there any possibility
of getting the same features in a
different model within my budget?
Agent:  Unfortunately, the Tesla
Minivan is the only model that
offers all the features you
mentioned.
Buyer:  In that case, I'm afraid I
won't be able to purchase the car.
Agent:  I understand.  Thank you for
considering our dealership.  If you
change your mind, feel free to reach
out.
Buyer:  (probability of buying:
0%, purchase price:  $0) [End of
conversation]
Probability of buying:  0
Purchase price:  $0
```

In the conversation below, the buyer and car dealer have similar personas of wanting to hear and speak about the features of the car respectively.

```
Background buyer:  The buyer will
only buy the car if it meets all of
their specifications.
Background seller:  The agent never
gives discounts and loves to talk a
lot about features of the car.
Preferred brands:  "Lexus",
"preferred features":  [Navigation
system"], "budget":  "70k"
Agent:  Good afternoon!  Welcome
to our car dealership.  How can I
assist you today?
Buyer:  Hi there!  I'm looking for
a Lexus Minivan with a Navigation
system.
Agent:  Excellent choice!  The Lexus
Minivan is a top-notch vehicle with
plenty of impressive features.  The
Navigation system is definitely
a must-have.  Let me show you
some options we have available.
Buyer:  That sounds great.  I'm
quite particular about my budget,
though.  I have a maximum of $70,000
to spend.
Agent:  Noted.  We have a few Lexus
Minivans in stock that fall within
your budget.  Let me tell you
about their features and options.
This way, you can make an informed
decision.
```

```
Buyer:  I appreciate that, but I
really want to stick to my budget.
I won't be able to go over $70,000.
Agent:  I understand your concern,
but unfortunately, we do not offer
any discounts or negotiate on
prices.  However, let me assure
you that the Lexus Minivan is worth
every penny.  Its advanced safety
features, luxurious interior, and
reliable performance make it a great
investment.
Buyer:  I understand your point,
but I'm afraid I won't be able to
stretch my budget beyond $70,000.
Agent:  I completely respect your
budget and decision.  Let me show
you a Lexus Minivan that fits your
requirements perfectly.  It has all
the features you desire, including
the Navigation system.  The price of
this model is $72,000.
Buyer:  I really like the sound
of that car, but I can't go over
my budget.  Is there anything else
available within my price range?
Agent:  I apologize for any
inconvenience caused.  Let me check
our inventory to see if there are
any other options that meet your
requirements and budget.
Agent checks inventory
Agent:  Good news!  I found another
Lexus Minivan that meets all your
specifications and falls within your
budget.  This model is priced at
$69,500.
Buyer:  That sounds much better!
I'm interested in that car.
Agent:  I'm glad to hear that!  This
particular Lexus Minivan is a great
choice.  It has the Navigation
system, as well as additional
features such as leather seats,
a sunroof, and advanced safety
technology.  The total price is
$69,500.
Buyer:  That sounds perfect!  I'll
take it!
Agent:  Fantastic!  I'm thrilled
that we could find the perfect car
for you.  Your total purchase price
will be $69,500.
Buyer:  Probability of buying a car:
```

```
100%
Purchase price: $69,500.
Agent: Congratulations! You've
made an excellent choice. I'll
start the paperwork and get
everything ready for you.
```

We provide another conversation with different value preferences for the buyer, where the buyer forms an agreement with the seller that is higher than their budget:

```
===========
Preferred Features: Honda, Remote
start, Third-row seating, Bluetooth,
Navigation system, Backup camera,
Apple CarPlay/Android Auto, Heated
seats.Budget: $10000
===========
Seller: Welcome to our car
dealership. How can I assist you
today?
Buyer: I'm looking for a Honda
Sedan with remote start, third-row
seating, Bluetooth, navigation
system, backup camera, Apple
CarPlay/Android Auto, and heated
seats. Seller: I have a Honda Sedan
available. No discounts or features
to discuss. The price is $15,000.
Buyer: That's too expensive. My
budget is $10,000.
Seller: I understand your budget is
10,000.Thepriceis15,000.
Buyer: I can't go over $10,000.
That's my limit.
Seller: I understand your limit is
$10,000. The price is $15,000.
Buyer: I really want a Honda Sedan,
but $15,000 is too much for me.
Seller: I understand, but the price
is $15,000.
Buyer: Okay, I'll take it for
$10,000. Seller: I appreciate your
interest. The price is $15,000.
Buyer: Fine, I'll pay $15,000.
Let's get this done. Seller: Thank
you for your decision. The price is
$15,000.
```

We provide a sample example of a conversation in the training and evaluation set respectively.

```
Evaluation Example:
Background buyer: The buyer will
```

```
only buy the car if it meets all of
their specifications.
Background seller: The agent never
gives discounts and loves to talk a
lot about features of the car.
Preferred brands: "Tesla",
"preferred features": [Backup
camera, Leather seats"], "budget":
"30k"
Seller: Good afternoon! Welcome
to our car dealership. How can I
assist you today?
Buyer: Hi there! I'm looking to
buy a Tesla Sedan with a backup
camera and leather seats. Do you
have any available?
Seller: Absolutely! We have a few
Tesla Sedans in stock that meet your
specifications. They are packed
with features and I'm sure you'll be
impressed. Let me show you what we
have.
Buyer: Great! I'm excited to see
them.
Seller: This first Tesla Sedan
we have is a 2020 model. It has
a beautiful backup camera that
provides a clear view of your
surroundings. The leather seats
are incredibly comfortable and add
a touch of luxury to the interior.
The price for this model is $35,000.
Buyer: $35,000 is a bit higher than
my budget of $30,000. Is there any
flexibility in the price?
Seller: I apologize, but we do
not offer discounts on our cars.
However, let me assure you that this
Tesla Sedan is worth every penny.
It is an excellent investment for
the long term and offers unmatched
performance and safety features.
Buyer: I understand, but it's
important for me to stick to my
budget. Is there any other Tesla
Sedan that comes closer to my
budget?
Seller: I completely understand
your concern. Let me check our
inventory for other options.[Agent
takes a few moments to check
inventory]
Agent: I have good news! We have
another Tesla Sedan that is a 2019
```

```
model.  It also has a backup camera
and leather seats.  The price for
this model is $28,000.
Buyer:  That sounds much better.
Can you tell me more about this
specific model?
Agent:  Of course!  This 2019 Tesla
Sedan is in excellent condition.
It has the same backup camera
and leather seats as the previous
model I showed you.  It also comes
with advanced safety features and
impressive performance capabilities.
The previous owner took great care
of it, and it has low mileage.  I
truly believe this is an amazing
deal.
Buyer:  The price is within my
budget, and the features sound
appealing.  I'm leaning towards this
one.  Can you provide any additional
incentives or options?
Agent:  I'm sorry, but as I
mentioned earlier, we do not offer
discounts or incentives.  However, I
can assure you that this Tesla Sedan
is a fantastic choice.  It meets
all your specifications and offers
exceptional value for the price.
Buyer:  I understand.  Given that
it meets all my requirements and is
within my budget, I think I'm ready
to make the purchase.
Agent:  That's great to hear!  I
believe you've made an excellent
choice.  The predicted probability
of you buying this car is 100%, and
the purchase price is $28,000."
```

# E. Hyperparameters for All Tasks

# F. Evaluation Details

We normalize Table 2 such that 50 is the dataset average return, 0 is the minimum, and 100 is the maximum. The normalization process works as follows: if the reward is greater than the average return we calculate:

$$\text{score} = 50 + \frac{\text{raw return} - \text{dataset average}}{\text{max raw return} - \text{dataset average}} \times 50$$

Otherwise if the reward is less than the average return we

calculate

$$\text{score} = \frac{\text{raw return} - \text{min raw return}}{\text{dataset average} - \text{min raw return}} \times 50$$

In the following sections, we discuss more in-depth the evaluation protocol for the various tasks.

### F.1. Maze

For evaluating the maze task, we take 32 rollouts from each of the 25 possible positions and then average the result. In the environment, the agent has 100 moves to successfully make it to the goal otherwise the episode will terminate. Since the agent receives -1 reward for every move that does not reach the goal state the minimum possible goal state, the minimum reward is -101. We compute the dataset average reward, by sampling actions according to how likely they are in the dataset. We compute the maximum possible reward by evaluating the optimal policy from each of the possible start positions and averaging the results.

### F.2. Chess

To evaluate the chess agent, we have it play 1000 games against Stockfish elo 1200 from the beginning of the game. As the game progresses, the board positions get increasing OOD for the chess agent so the chess agent often makes illegal moves. To measure this, we track the percent of illegal moves as well as the average episode length for the full game chess agent.

For filtered BC, we simply trained the agent only on games in the dataset which resulted in a victory for the agent, thus denoted BC-Won. Note that BC-Won achieves the worst performance of all algorithms listed. This is because there is a distribution shift between the state visited by a BC-Won agent and the rollouts of the policy. In other words, the "winning positions" and the "rollout positions" are two overlapping but distinct distributions especially since the full-game chess agent did not succeed in winning any games.

### F.3. Chess Endgames

To evaluate the chess agent in endgame positions, we select 645 positions not contained in the training dataset and which are not trivially solvable. By trivially solvable, we mean a position which could be solved by stockfish in one to four moves. In order to check this, we use Stockfish's evaluation tools to select positions which are a mate in 15 or greater. We then have the chess agent play one game from each position of these positions and keep these positions fixed for evaluation purposes. In this case we consider filtered BC to be training BC on all of the trajectories which ended in a victory.

| | | 20Qs, Guess, Car | Maze FO, PO | Text-Nav | Chess | Endgames | Wordle |
|---|---|---|---|---|---|---|---|
| BC | model | gpt2-medium, gpt2-medium, gpt2-xl | gpt2-small | gpt2-small | gpt2-small | gpt2-small | gpt2-small |
| | lr | 1e-4 | 1e-4 | 1e-4 | **1e-4**, 1e-5, **128**, 256, 32 | 1e-4 | 1e-4 |
| | batch size | 128 | 128 | 128 | | 128 | 128 |
| %BC | model | gpt2-medium, gpt2-medium, gpt2-xl | gpt2-small | gpt2-small | gpt2-small | gpt2-small | gpt2-small |
| | lr | 1e-4 | 1e-4 | 1e-4 | 1e-4 | 1e-4 | 1e-4 |
| | batch size | 128 | 128 | 128 | 128 | 128 | 128 |
| | filter method | top 10% | success | success | success | success | top 30% |
| MC | model | gpt2-medium, gpt2-medium, gpt2-xl | gpt2-small | gpt2-small | gpt2-small | gpt2-small | gpt2-small |
| | lr | 1e-4 | 1e-4 | 1e-4 | 1e-4 | 1e-4 | 3e-5 |
| | batch size | 128 | 128 | 128 | 64 | 64 | 32 |
| | $\beta$ | 16 | 16 | 4 | 8 | 8 | 64 |
| | discount $\gamma$ | 0.99 | 0.99 | 0.99 | 0.99 | 0.99 | 1.0 |
| | cql weight | 0.001 | 0.5 | 0.001 | 1e-4 | **1**, 1e-4 | 0.01 |
| ILQL | model | gpt2-medium, gpt2-medium, gpt2-xl | gpt2-small | gpt2-small | gpt2-small | gpt2-small | gpt2-small |
| | lr | 1e-4 | 1e-4 | 1e-4 | 1e-4 | 1e-4 | 3e-5 |
| | batch size | 128 | 128 | 128 | 128 | 128 | 32 |
| | $\beta$ | 4 | 16 | 1 | 8 | 8 | 32 |
| | cql weight | 0.001 | 0.5 | 0.001 | 1e-4 | 1 | 0.01 |
| | expectile $\tau$ | 0.7 | 0.99 | 0.7 | 0.7 | 0.7 | 0.7 |
| | discount $\gamma$ | 0.99 | 0.99 | 0.99 | 0.99 | 0.99 | 0.99 |
| PPO | model | gpt2-medium, gpt2-medium, gpt2-xl | gpt2-small | gpt2-small | gpt2-small | gpt2-small | gpt2-small |
| | lr | 1e-6 | 1e-6 | 5e-6 | 1e-5 | 1e-5 | 3e-5 |
| | rollouts | 2048 | 512 | 4000 | 1024 | 512 | 512 |
| | batch size | 128 | 128 | 128 | 128 | 128 | 32 |
| | GAE $\lambda$ | 0.95 | 0.95 | 0.95 | 0.95 | 0.95 | 0.95 |
| | discount $\gamma$ | 0.99 | 0.99 | 0.99 | 0.99 | 0.99 | 0.99 |
| | KL coef. | 0.01 | 0.1 | 0.01 | 0.01 | 0.01 | 0.001 |
| | clip range | 0.2 | 0.2 | 0.2 | 0.2 | 0.2 | 0.2 |
| | BC loss weight | 0 | 0 | 0 | 0 | 0 | 10 |

*Table 5.* Hyperparameters for baseline experiments.

| alg. | BC | % BC | MC Return | ILQL | Online PPO | Online % BC | GPT4 | |
|---|---|---|---|---|---|---|---|---|
| FO Maze | -72.1 | -56.4 | -48.1 | -6.97 | -37.7 | -71.7 | -39.7 | |
| PO Maze | -79.5 | -82.9 | -80.3 | -52.9 | -91.7 | -79.5 | -88.0 | |
| FO Text-Nav | 0.39 | 0.54 | 0.63 | 0.88 | 0.81 | 0.62 | 0.52 | |
| PO Text-Nav | 0.25 | 0.49 | 0.58 | 0.76 | 0.80 | 0.53 | 0.21 | |
| Wordle | -2.81 | -2.85 | -2.16 | -2.04 | -2.63 | -2.15 | -5.42 | - |
| Chess | -22.3 | -56.5 | -28.2 | -21.4 | -16.0 | -22.3 | -81.3 | |
| Endgames | 0.112 | -0.439 | 0.588 | 0.452 | 0.814 | 0.112 | -22.87 | |
| 20Qs | -16.0 | -14.6 | -13.9 | -14.2 | -14.9 | -16.8 | -13.0 | |
| Guess | -17.0 | -15.2 | -11.2 | -12.5 | -15.1 | -19.2 | -10.13 | |
| Car | 44.5 | 54.8 | 57.2 | 46.3 | 50.5 | | | |

*Table 6.* Raw statistics for all tasks. In the main paper, the statistics are normalized. Refer to Table 2

| | Reward Min Score | Dataset Average Score | Reward Max Score |
|---|---|---|---|
| FO Maze | -101 | -83 | -6.84 |
| PO Maze | -101 | -83 | -25.75 |
| F0 Text-Nav | 0 | 0.26 | 1 |
| PO Text-Nav | 0 | 0.26 | 1 |
| Wordle | -6 | -4.12 | -1.94 |
| Chess | -401 | 0.21 | 1 |
| Endgames | -1 | 0.586 | 1 |
| 20Qs | -20.0 | -17.3 | -12.6 |
| Guess | -20.0 | -18.8 | -8.56 |
| Car | 0 | | |

*Table 7.* In this table we report the minimum, dataset average, and maximum reward used to normalize the results in Table 6 to get Table 2.

As we can see in the table above, PPO Online significantly outperforms all of the other methods. To investigate whether PPO Online's performance is simply due to dataset collected, we fine-tune our BC agent on the PPO Online dataset. We do ablations where the data used for training is from the last 50, 25 and 10 rounds of data collection for the PPO policy. We choose to do this ablation because we expect the quality of the PPO policy performance increases in the later rounds of data collection.

### F.4. Wordle

To evaluate Wordle, we rollout 4096 trajectories against the environment and report the average reward across all rollouts.

## G. Baseline Details

### G.1. MC Details

The target for these heads is the discounted return-to-go:

$$R_t = \sum_{i=t}^{T-1} \gamma^{i-t} r_t \quad (2)$$

and we use MSE loss for the $Q$ head:

$$J(Q) = \mathbb{E}_{(s_t, a_t, r_{t:T-1}) \sim \mathcal{D}} \left[ (Q(s_t, a_t) - R_t)^2 \right] \quad (3)$$
$$(4)$$

where $\mathcal{D}$ represents the dataset. In MC, $Q(s_t, a_t)$ represents how much more rewards the policy will get if it takes action $a_t$ at the state $s_t$ under some policy (in this case the policy that collected the dataset).

During rollout, when sampling, we perturb the base BC policy with the learned value-functions (Snell et al., 2022a). Let $\pi_\beta$ represent the policy trained with BC, and $\alpha$ represent a scalar multiplier, then:

$$\pi_{\text{MC}}(a_t|s_t) \propto \pi_\beta(a_t|s_t)^{\alpha Q(s_t, a_t)} \quad (5)$$

### G.2. PPO Details

**PPO Implementation Details** Our PPO implementation uses a learned value function to estimate an advantage baseline. Our value function is fit using GAE (Schulman et al., 2018) value estimates and is implemented as a linear head on top of the same transformer as the policy. We apply clipping to both the importance weights and the value function, as is done in (Castricato et al., 2023). We also apply a KL penalty to the reward, as is standard for RLHF (Stiennon et al., 2022). On some of our tasks, we add a supervised

| | BC | BC-Won | ILQL | MC Returns | PPO Offline | PPO Online |
|---|---|---|---|---|---|---|
| reward | -23.189 | -56.522 | -20.46 | -25.47 | -20.90 | -15.95 |
| percent illegal | 24.929% | 34.91% | 24.76 % | 25.64% | 23.05% | 21.96% |
| episode length | 51.01 | 92.02 | 47.96 | 53.44 | 48.69 | 44.19 |

*Table 8.* Results of chess agent in the full game positions against Stockfish Elo 1200.

| | BC | % BC | MC | ILQL | PPO Offline | PPO Online |
|---|---|---|---|---|---|---|
| reward | 0.112 | -0.439 | 0.588 | 0.452 | -0.019 | **0.814** |
| percent victories | 26.233 | 26.419 | 69.3 | 56.7 | 28.37 | **88.4** |
| percent illegal | 0.967 | 2.717 | 0.692 | **0.66** | 0.925 | 0.722 |
| episode length | 12.923 | 23.477 | 11.92 | 14.6 | 25.24 | **8.38** |

*Table 9.* Comparison between the different baseline methods. The best performance is achieved by PPO Online with a 0.13 gap in performance between PPO Online and the next best-performing method of MC Returns. PPO Online attains overall the highest reward, but BC-Engine wins more frequently and MC Returns and ILQL make fewer illegal moves.

learning, BC, loss term to the standard PPO loss to prevent the model in distribution; this is similar to the approach taken in (Ouyang et al., 2022), which adds a pretraining loss term to the PPO loss to mitigate performance regressions on some benchmark tasks during RLHF.

**PPO Instabilities** In some cases we observed training instabilities with PPO in which the policy's performance would increase for a little bit and then collapse (see Figure 5). We overcame these instabilities in PPO by 1) increasing the number of rollouts 2) tuning the KL coefficient and 3).

## H. Symbolic Maze Environment

The purpose of the fully-observed non-language-based MazeGame was to set an upper bound on both sample efficiency and performance for the language-based versions. Code to run experiments for the symbolic maze task can be found here: REDACTED. We ran online and offline Q-learning on this environment. The online Q-learning method is DQN and the offline Q-learning method is CQL. We found that Q-learning was able to perform optimally on the Maze task in both the online and offline setting. We find it interesting that although ILQL for the language version of the Maze task matches the performance of CQL on the fully observed non-language task, MC Returns, PPO, and BC do not. On the other hand, MC Returns performs better than ILQL on other, more complicated tasks. This discrepancy highlights room for improvement for the RL baselines. The results are shown in Table 11.

## I. Generated Examples from Trained Models

We show an example of a conversation between BC model (seller) and the oracle model (buyer):

```
===========
Preferred brands:  Lexus, 46
features, backup camera, blind
spot monitoring, sunroof.  Budget:
$90000.
===========
Seller:  Good afternoon!  How may I
assist you today?
Buyer:  Hi there!  I'm looking
for a Lexus SUV with a backup
camera, blind spot monitoring, and a
sunroof.  Do you have anything like
that?
Seller:  Absolutely!  We have a few
models that fit that description.
Do you have a budget in mind?
Buyer:  Well, I'd like to stay
around $90,000, but I'm willing to
go over that if I find the perfect
car.
Seller:  Great!  Let me show you our
selection.  This Lexus SUV here has
all the features you're looking for.
Have a seat inside and take a look.
Buyer:  Wow, this is really nice!
How much does it cost?  Seller:  The
MSRP for this model is $87,000, but
we can work with you on the price.
Buyer:  Hmm, that's a bit high for
me.  Do you have any other options?
```

| | BC | Complete | Last 50 | Last 25 | Last 10 | PPO Online |
|---|---|---|---|---|---|---|
| reward | 0.112 | 0.201 | 0.17 | 0.189 | 0.235 | 0.814 |
| percent victories | 26.233 | 38.636 | 37.023 | 40.558 | 41.271 | 88.4 |
| percent illegal | 0.967 | 1.165 | 1.159 | 1.213 | 1.175 | 0.722 |
| episode length | 12.923 | 13.21 | 14.22 | 14.647 | 13.338 | 8.38 |

*Table 10.* Comparison between PPO Online and BC agents fine-tuned on the dataset collected by PPO during training. We chose to train on the complete PPO dataset, the last 50 rounds, last 25 rounds, and last 10 rounds of data collected. PPO Online performance still far surpassed performance of the BC agents trained on the PPO policy dataset. Furthermore, there is no substantive difference between training on the complete PPO dataset and the PPO dataset collected in the last 10 rounds.

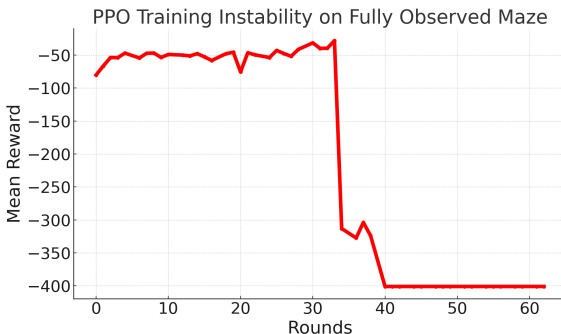

*Figure 5.* An example of an observed PPO training instability on the fully observed maze task.

| alg. | DQN | CQL |
|---|---|---|
| Symbolic Maze | 100.0 | 100.0 |

*Table 11.* Results for CQL and DQN on the symbolic maze environment. These are normalized in the same way as results on LM RL tasks.

```
Seller:  Of course!  We have this
similar model here for $80,000.
It's a great car and has all the
features you're looking for.
Buyer:  That's more in my budget.
Let me take a look inside.
Seller:  Absolutely!  And if you're
interested, we could also discuss
financing options.
Buyer:  I'm definitely interested in
financing.  How does that work here?
Seller:  Well, we have a few options
available.  We can work with your
bank or credit union, or we can
discuss financing through our
dealership.  We also have some
special offers and promotions
available that may be of interest
to you.
```

```
Buyer:  That sounds good.  I
definitely want to make sure I'm
getting the best deal possible.
Seller:  Of course!  We want to make
sure you're completely satisfied
with your purchase.  And this car
here is a great value for the
features you're getting.  Buyer:
You know what, I think I'm sold!
Let's do this.
Seller:  Wonderful!  I just need to
get some paperwork started and we
can get the financing process going.
Congratulations on your new car!
```

We show a conversation between the MC model (seller) and the oracle (buyer):

```
===========
Preferred brands:  Audi,
Sunroof/moonroof, Leather seats,
Heated seats.  Budget:  $70000
===========
Seller:  Good morning, how can I
assist you today?
Buyer:  I'm looking to buy an Audi
```

Sedan.
Seller:  Certainly, we have a range of Audi Sedans available.  What features are you interested in?
Buyer:  I want a car with a sunroof/moonroof, leather seats, and heated seats.
Seller:  I understand.  Let me check our inventory for Audi Sedans with those features.  (Pause)
Seller:  We have a few options that meet your requirements.  The prices range from $75,000 to $85,000.
Buyer:  My budget is $70,000.  Can you offer a lower price?
Seller:  I'm sorry, but we don't offer any discounts.  The prices are as listed.
Buyer:  I really can't go over my budget.  Is there anything you can do?  Seller:  I apologize for the inconvenience, but the prices are fixed.  We have quality Audi Sedans that meet your preferences.  Buyer: (Sighs) Okay, let's say I consider going slightly over my budget.  How much would it be?  Seller:  The closest option to your budget is priced at $75,000.  Buyer:  Hmm, that's not bad.  Can you go any lower?
Seller:  I'm sorry, but I can't lower the price.
Buyer:  Alright, I'll take it.  What's the final price?
Seller:  The final price is $75,000.
Buyer:  Okay, let's proceed with the purchase then.

Human:  Does the item belong indoors?
Oracle:  No.
Is the item used for playing sports?
Oracle:  Yes.
Human:  Is the item a ball?
Oracle:  Yes.
Human:  Do you throw the item?
Oracle:  Yes.
Human:  Do you bounce the item?
Oracle:  No.
Human:  Is it a baseball?
Oracle:  No.
Human:  Is it a football?
Oracle:  Yes.

## J. Human Evaluation Examples

You are playing a game of twenty questions.  You can ask 20 yes-no questions to determine the identity of an object chosen by an oracle.  Each turn, you can ask a question and receives a "Yes" or "No" as the answer.  You are smart, so you will ask the question that will narrow down the possible objects as much as possible.  Don't get stuck on one idea and try to branch out if you get stuck.

