# OpenReview forum: "LMRL Gym: Benchmarks for Multi-Turn Reinforcement Learning with Language Models"
_ICML.cc/2025/Conference — ICML 2025 poster_

### Official Review · Reviewer_MQ7V · 2025-02-15

**Overall Recommendation:** 3

**Summary:**

This paper introduces the LMRL-Gym benchmark for evaluating multi-turn RL for LLMs, together with an open-source research framework. The proposed benchmark consists of 3 Interactive Dialogue tasks and 5 RL Capability tests, which require multiple rounds of language interaction and cover a range of tasks in open-ended dialogue and text games.

**Claims And Evidence:**

Yes.

**Essential References Not Discussed:**

N/A

**Experimental Designs Or Analyses:**

Yes.

**Methods And Evaluation Criteria:**

N/A

**Other Comments Or Suggestions:**

N/A

**Other Strengths And Weaknesses:**

## Strengths
1. The proposal of an interactive simulator for benchmarking RL performance on multi-turn tasks is novel and important
2. The paper is clearly presented with proper examples and figures.
3. The release of data and toolkit can help the research community


## Weaknesses
1. How does the dialogue tasks in the proposed benchmark differ from classical multi-turn task-oriented dialog dataset, such as MultiWOZ [1]
2. Experiments are only performed on GPT-2 models, casting doubts on the scalability of the results and the validity of the proposed benchmarks on larger models

***
[1] https://github.com/budzianowski/multiwoz

**Questions For Authors:**

1. What is the benefit of formulating language generation tasks as a partially observable Markov decision process rather than the standard fully observable MDP?
2. What are the benefits of the proposed tasks over math problem solving or code generation in testing RL training on LLM agents?

**Relation To Broader Scientific Literature:**

A benchmark for LM's ability on multi-turn dialog is well needed for the research community.

**Theoretical Claims:**

N/A

---

> ### Author Rebuttal · Authors · 2025-04-01
>
> We thank the reviewer for their feedback and questions. We would like to answer the questions in your review as follows:
>
> 1. "How does the dialogue tasks in the proposed benchmark differ from classical multi-turn task-oriented dialog dataset, such as MultiWOZ?"
>
> The goal of our paper is to present a benchmark that applies RL algorithms to multi-turn tasks, specifically to perform goal-directed dialogue. To this end, we provide (1) online simulators and offline datasets for a suite of 7 text-based strategy games and dialogue tasks (2) methodology to create simulators for offline evaluation, online RL training, and computing rewards (3) a research framework and toolkit for researchers and practitioners to get started with multi-turn RL for LLMs (focusing on both online & offline RL), which includes implementations of PPO, ILQL, and several baseline methods. MultiWOZ primarily focuses on dialog and does not provide the ability to test algorithms on specific RL capabilities including trajectory stitching, credit assignment and partial observability. Additionally, they do not focus on both online and offline reinforcement training algorithms for LLMs. Please refer to response to Reviewer 7KJe for detailed discussion on other related works.
>
> 2. "Experiments are only performed on GPT-2 models, casting doubts on the scalability of the results and the validity of the proposed benchmarks on larger models"
> We focus primarily on GPT2-models as there is a good understanding of its capabilities and broad usage in research for establishing baselines [3] compared to newer, similar sized models. We would like to highlight several recent works [1, 2, 4, 5] that have also used GPT2 for fine-tuning. Our paper focuses on providing a framework that can be easily adapted to various models and the development of further algorithms for RL fine tuning for LLMs, and we hope to see further works that iterate upon other models.
>
> [1] Hicke, Y., Masand, A., Guo, W., & Gangavarapu, T. (2024). Assessing the efficacy of large language models in generating accurate teacher responses. Proceedings of the 62nd Annual Meeting of the Association for Computational Linguistics (ACL 2024). arXiv preprint arXiv:2401.12345.
>
> [2] Hong, J., Dragan, A. D., & Levine, S. (2024). Q-SFT: Q-learning for language models via supervised fine-tuning. Proceedings of the 42nd International Conference on Machine Learning (ICML 2024). arXiv preprint arXiv:2403.01512.
>
> [3] Radford, A., Wu, J., Child, R., Luan, D., Amodei, D., Sutskever, I., et al. (2019). Language models are unsupervised multitask learners. OpenAI Blog, 1(8), 9. 4o mini
>
> [4] Zhou, R., Du, S. S., & Li, B. (2024). Reflect-RL: Two-player online RL fine-tuning for LMs. Proceedings of the 62nd Annual Meeting of the Association for Computational Linguistics (ACL 2024). arXiv preprint arXiv:2401.12345.
>
> [5] Zhou, Y., Zanette, A., Pan, J., Levine, S., & Kumar, A. (2024). ArCHer: Training Language Model Agents via Hierarchical Multi-Turn RL. Proceedings of the 41st International Conference on Machine Learning (ICML 2024), 235, 62178–62209. arXiv preprint arXiv:2402.19446.
>
> 3. "What is the benefit of formulating language generation tasks as a partially observable Markov decision process rather than the standard fully observable MDP?"
>
> We formulate language as a POMDP, as the true state of the world is not completely represented in text form. In language tasks, the state consists of the entire history of tokens, and an agent may need to examine this entire context to infer the correct state. The mental states of a speaker in a dialogue (e.g., whether the buyer is impatient in a selling task), previously observed facts in a guessing game, and other hidden variables might induce partial observability.
>
> 4. "What are the benefits of the proposed tasks over math problem solving or code generation in testing RL training on LLM agents?"
>
> We acknowledge that math problems and code generation are interesting tasks to test the capabilities of RL-LLM algorithms. However, we chose our text game tasks as they are similar in nature to traditional RL tasks, but with a twist of including language to test how this impacts the performance of RL algorithms for LLMs. This allows us to isolate traditional RL issues such as credit assignment and trajectory stitching but in a language-based setting. To that end we have designed five tasks as RL Capability Tests, which are text games designed to isolate specific capabilities of RL training as shown in Figure 4. As seen, these text-games do not test all of the capabilities of RL together, which is only possible through the dialogue-based tasks. Please refer to response to Reviewer RuRY on more clarification on our choice of tasks for the benchmark.

---

### Official Review · Reviewer_gYws · 2025-02-16

**Overall Recommendation:** 4

**Summary:**

This paper introduces LMRL-Gym, a benchmark framework for evaluating multi-turn reinforcement learning with language models, consisting of 8 tasks divided into interactive dialogue tasks and RL capability tests. The framework includes implementations of several baseline methods (PPO, ILQL, behavior cloning) and evaluates them against GPT-4. The authors also provide a comprehensive toolkit for researchers to develop and evaluate RL algorithms for LLMs.

**Claims And Evidence:**

The paper's main claims face two significant issues:

Novelty Claims:


Claims to be the first comprehensive benchmark for multi-turn RL with LLMs
However, numerous existing frameworks already address similar challenges (MineCraft, StarCraft, BabyAI)
The paper fails to acknowledge or compare against these established works
Many claimed contributions are incremental combinations of existing approaches


Empirical Support:


The experimental results don't convincingly demonstrate the benchmark's value
Limited analysis of why different algorithms perform differently
No clear evidence that the benchmark captures important aspects of LLM capabilities
Results with outdated models limit the practical relevance of findings

**Essential References Not Discussed:**

Several critical works are missing that fundamentally challenge the paper's claimed contributions:

LLM Agent Environments and Benchmarks:


"MineDojo: Building Open-Ended Embodied Agents with Internet-Scale Knowledge" (NeurIPS 2022) - Demonstrates complex multi-turn LLM interactions in Minecraft
"BabyAI: A Platform to Study the Sample Efficiency of Grounded Language Learning" (ICLR 2019) - Provides a similar framework for language-based RL
"VoyageAI: An Open-Ended Embodied Agent with Large Language Models" (2023) - Shows advanced multi-turn interaction capabilities

**Ethical Review Concerns:**

Not related to any ethics topic.

**Ethics Expertise Needed:**

["Other expertise"]

**Experimental Designs Or Analyses:**

Two critical limitations in experimental design:

Baseline Comparisons:


Limited comparison with existing benchmarks
No ablation studies on task design choices
Insufficient analysis of failure cases
Missing comparison with recent prompt-based methods


Model Selection:


Relies on outdated models and architectures
No experiments with more recent, efficient models
Limited scale of experiments
Missing analysis of computational requirements

**Methods And Evaluation Criteria:**

Two major concerns with the methodology:

Task Design Issues:


The 8 tasks appear to be simple combinations of existing work
No clear justification for why these specific tasks were chosen
Limited novelty in task design and implementation
Tasks may not be challenging enough for modern LLMs


Technical Implementation:


Uses outdated GPT-2 models instead of modern alternatives (Phi, Qwen)
Limited model scale compared to current standards
No utilization of recent advances in efficient fine-tuning
Benchmark design doesn't account for latest developments in LLM capabilities

**Other Comments Or Suggestions:**

The paper requires substantial revision in two key areas:

Technical Development: Update the experimental framework to include modern efficient models (Phi, Qwen2-0.5/1.5B, TinyLlama-1.1B), provide thorough scaling analysis, and move critical experimental results from supplementary materials to the main text. The analysis should focus particularly on what differentiates multi-turn from single-turn performance and why certain models perform differently across tasks.


Literature Integration: Thoroughly engage with existing LLM agent literature, particularly works on multi-turn interaction in environments like Minecraft，StarCraft2，Overcook and BabyAI. The paper needs to clearly articulate its unique contribution in light of these works and provide detailed comparisons with existing benchmarks.

**Other Strengths And Weaknesses:**

The paper's main strength lies in its comprehensive implementation framework, providing a complete pipeline from environment setup to model evaluation, with clear documentation and reproducible experiments. The integration of multiple baseline methods (PPO, ILQL, BC) makes it potentially useful for practitioners entering the field.

However, the work suffers from two fundamental weaknesses: First, the technical execution relies heavily on outdated models and lacks thorough analysis, particularly regarding the impact of model scale and multi-turn interactions. Second, the research contribution is significantly diminished by inadequate engagement with existing work, especially in relation to established LLM agent benchmarks and environments.

**Questions For Authors:**

Model Selection and Analysis:


Why did the paper primarily use older models like GPT-2 when recent efficient models (e.g., Phi-2, Qwen2-0.5/1.5B，TinyLlama-1.1B) are readily available with similar computational requirements?
The performance comparison shows GPT-4 performing poorly on some tasks (e.g., Chess, Endgames) but excelling at others (e.g., 20Qs, Guess). What explains this discrepancy? A deeper analysis of model capabilities and task characteristics would be valuable.


Experimental Depth:


The paper lacks analysis comparing multi-turn versus single-turn performance. How do the benefits of multi-turn interaction manifest in your tasks?
How does model size impact performance across different tasks? The current experiments don't explore this important dimension.
Why were the detailed experimental results moved to supplementary materials rather than being presented in the main text, given their importance to your claims?


Environment Design:


What motivated the selection of these specific 8 environments? How do they provide unique value compared to existing environments in both LLM agent research (e.g., MineCraft, StarCraft) and traditional RL?
The paper mentions these tasks test different capabilities, but how do you validate that they actually measure the intended capabilities distinctly?
How do you ensure these environments pose meaningful challenges for modern LLMs while remaining computationally tractable?


Methodology and Results:


Given the supplementary materials contain important analyses and experimental details, why weren't key insights from these analyses highlighted in the main paper?
How do your hyperparameter choices and training procedures compare to those used in similar LLM fine-tuning work?
What specific challenges did you encounter in training models on these environments that might inform future research?

**Relation To Broader Scientific Literature:**

Two major gaps in literature coverage:

Missing Related Work:


No discussion of major LLM agent works (MineCraft, StarCraft, Overcooked)
Ignores significant work on prompt-based RL methods
Missing comparison with BabyAI and similar frameworks
Limited acknowledgment of recent advances in LLM fine-tuning


Context and Positioning:


Fails to properly position the work within existing literature
Overstates novelty and contribution
Missing discussion of recent trends in LLM agent development
Limited connection to broader RL literature

**Theoretical Claims:**

No significant theoretical claims to verify, though the paper would benefit from more theoretical analysis of why certain algorithms perform better on specific tasks.

---

> ### Author Rebuttal · Authors · 2025-04-01
>
> We thank the reviewer for their feedback and suggestions for related works. We will be sure to the cite the works that you have noted, and move material from our paper to the Appendix / supplementary section. To address the questions in your review, we would like to provide responses to the following:
>
> 1. "Task Design Issues": We would like to clarify that each of the tasks in the benchmarks serve a different purpose, and we would like each task to isolate different RL capabilities to develop better algorithms. Please refer to #2 response to Reviewer RuRY
> 2. "Related Work": We will be sure to cite the related works you have noted. Please refer to #1 response to Reviewer gYws on discussion of related works.
> 3.  "Model Selection": Regarding model selection and usage of GPT-2, please refer to #1 response of Reviewer RuRY.
>
> Regarding your question on analysis of failure cases and our models, we would like to clarify the methodology with which we generated our datasets to train our simulators, and how we ensured high quality and consistency for these datasets. As shown in Figure 2, we train a simulator that serves as an “oracle” for the task, and hence does not require any capabilities of strategic reasoning, but provides signals to help the agent model learn. For example, the role of the oracle in the Twenty Questions task is to provide objective yes/no answers to questions about the object, and in Guess My City, to provide more open ended information about a query on the city. OpenAI’s GPT-3.5 has been shown to be able to generate reasonable questions and answers when used out of the box, which is why we leveraged it to collect our initial dataset. We have provided prompts that we use to generate the data to train our oracle models in our Appendix, and snippets below to show our thought process to maintain high accuracy.
>
> The method for collecting the dataset is as follows. For each conversation, we select uniformly at random from the above list the word that the oracle is answering question about. The oracle is an LLM (OpenAI’s GPT3.5) given the following prompt. In our prompts, we denote variables that we fill in with variable data with {{variable}}.
>
> Prompt: You are a question answering oracle. You will answer each question about an object with Yes or No. If the answer could be both, answer with the most typical scenario. Here’s a few examples:
>
> example 1:
>
> object: Computer
>
> question: Does the object use electricity?
>
> answer: Yes.
>
> explanation of answer: Computers need electricity to function. [...]
>
> Additionally, we have also validated the data from trained oracle models through human evaluation. We have also provide generated examples by both oracle models and trained agents in our Appendix. With respect to the Car Dealer task, we spent a considerable effort to ensure diversity in the responses of sellers, by providing different desired brands, features, classifications (i.e. car or truck), and budgets. We have provided samples of conversation between the oracle model and MC returns vs. oracle and the BC model in the Appendix.

---

> > ### Comment · Reviewer_gYws · 2025-04-02
> >
> > I strongly recommend testing at least one recent Small LLM (Qwen, Phi) in your environments. This test could be implemented with just one GPU in only one day, yet would significantly strengthen your paper's relevance and contribution.
> >
> > Please do not hesitate to do this, otherwise I cannot recommend acceptance and will lower my score.

---

> > > ### Author Response · Authors · 2025-04-09
> > >
> > > We thank the reviewer for their comment. As suggested, we fine tuned Qwen2.5-VL-3B-Instruct for the 20 Questions task, as this model is reported to be better for chit-that and instruction following. We report the following results:
> > >
> > > | **Task**       | --- | BC  | MC  | BC% | PPO |
> > > |----------------|-----|-----|-----|-----|-----|
> > > | **20 Questions** |     | 62.4 | 87.4 | 86.1 | 76.3
> > >
> > > Compared to results for gpt2 in Table 2, Qwen2.5-3B demonstrates higher accuracy for BC, %BC and Online PPO, and similar accuracy for MC. We hypothesize that this might be due to its ability to demonstrate strategic behavior and better reasoning capabilities without a lot of training data. We will add Qwen2.5-3B to our LMRL-Gym repository for others to train with and report further results for the other domains in our paper, as per your suggestion. As it takes several days to train ILQL, we have not reported this result in the table but will do so for the final paper.

---

### Official Review · Reviewer_7KJe · 2025-03-14

**Overall Recommendation:** 2

**Summary:**

This paper introduces LMRL-Gym, a benchmark for evaluating reinforcement learning algorithms for multi-turn generation of large language models (LLMs). LMRL-Gym provides 3 interactive dialogue tasks and 5 RL capability tasks. More specifically, interactive dialogue tasks include 20Qs (Twenty Questions), Guess (Guess My City), Car Dealer. RL capability tasks include Maze, Text-Nav, Wordle, Chess, and Endgames. Also, LMRL-Gym provides variants of behavior cloning, offline value-based RL (e.g., ILQL), and online RL (e.g., PPO) as baselines. Finally, this paper provides experiment results that evaluate baseline RL algorithms on the 8 tasks.

**Claims And Evidence:**

This paper aims to provide a benchmark for evaluating RL algorithms for multi-turn generation of LLMs. However, LMRL-Gym only provides two types of tasks like interactive dialogue tasks and test-based games. It seems rather restricted to be a general benchmark. Also, the diversity of the baseline algorithms seems limited, since it only includes BC, ILQL, and PPO.

**Essential References Not Discussed:**

This paper does not comprehensively discuss essential related works. There are many benchmarks for LLMs. Also, there are benchmarks specialized for multi-turn generation or RL fine-tuning. This paper seems to discuss only small part of the related works.

**Experimental Designs Or Analyses:**

This paper evaluates the baseline RL algorithms (BC, ILQL, and PPO) on the eight proposed tasks (20Qs, Guess, Car, Maze, Text-Nav, Wordle, Chess, and Endgams).

**Methods And Evaluation Criteria:**

Since this paper is a benchmark paper, there is no proposed method. By the way, for the purpose of baselines, this paper provides BC, ILQL, and PPO. However, the diversity of the baselines seems limited to assess the usefulness of the proposed benchmark.

**Other Comments Or Suggestions:**

C1. The term “RL capability tasks” seems rather unclear. It would be better to revise it to be more clear term.

C2. Figure 3 seems rather unclear. Please revise it.

**Other Strengths And Weaknesses:**

Other weaknesses:

W1. For the interactive dialogue tasks, this paper uses LLMs (i.e., GPT-3.5 and GPT-2). I am not sure that the quality of the generated data is sufficient to be used for a benchmark.

**Questions For Authors:**

Q1. Why do the authors use FLAN-T5-XL and GPT2-XL for generating interactive dialogue tasks? They seem rather small and old.

**Relation To Broader Scientific Literature:**

This paper introduces MLRL-Gym, a benchmark for evaluating RL algorithms for multi-turn generation of LLMs. Improving the multi-turn capability of LLMs with RL is one of important research topics. However, the proposed benchmark seems rather limited to be a representative benchmark for RL algorithm for multi-turn generation.

**Theoretical Claims:**

This paper is a benchmark paper. It does not present any proofs for theoretical claims.

---

> ### Author Rebuttal · Authors · 2025-04-01
>
> We thank the reviewer for their feedback and suggestions for clarity on the work. We will revise our figure and term accordingly as per your suggestion. We've addressed the main questions raised in your review by: (1) providing an extensive literature review of other popular benchmark papers (2) clarifying our methodology to generate data to train our simulators (discussed in Reviewer gYws response) (3) clarifying your question on use of FLAN-T5-XL and GPT2-XL (discussed in Reviewer RuRY and Reviewer MQ7V response).
>
> **Related Works**: In order to clarify the contribution of the paper, we provide comparison to popular related works in text games, interactive dialog tasks and offline RL.
>
> [1] Chevalier-Boisvert, M., Bahdanau, D., Lahlou, S., Willems, L., Saharia, C., Nguyen, T. H., & Bengio, Y. (2018). Babyai: A platform to study the sample efficiency of grounded language learning. arXiv preprint arXiv:1810.08272.
> - It is not a text-based representation, and instead a state is passed as a vector, and RL is trained on the state
> - This task cannot be easily used to evaluate RL/LLM tasks
>
> [2] Gontier, N., Rodriguez, P., Laradji, I., Vazquez, D., & Pal, C. (2023). Language Decision Transformers with Exponential Tilt for Interactive Text Environments. arXiv preprint arXiv:2302.05507.
> - Results indicate they may not have collected enough data for offline RL algorithms, as offline RL performs poorly [17, 18,19,20]
>
> [3] Hausknecht, Matthew, et al. "Interactive fiction games: A colossal adventure." Proceedings of the AAAI Conference on Artificial Intelligence. Vol. 34. No. 05. 2020. Introduces the Jericho Benchmark
> - Our smallest task includes a dataset of 1.25k trajectories. This dataset contains 590 trajectories. A large, diverse dataset is critical for testing offline RL [17, 18]
> - Our benchmark is not only text-games and using templates for interaction, we utilize free-form text generation and simulate human-AI interaction
>
> [4] Shridhar, M., Yuan, X., Côté, M. A., Bisk, Y., Trischler, A., & Hausknecht, M. (2020). Alfworld: Aligning text and embodied environments for interactive learning. arXiv preprint arXiv:2010.03768.
> - The work is similar to the TextWorld benchmark, but LMRL-Gym benchmark is a lot more than Text-Nav, and this is our simplest task mainly meant to test implementation and correctness (e.g. “unit test”)
> - LMRL-Gym has other text-games and dialogue tasks that are more complex and test a variety of RL Capabilities such as credit assignment, trajectory stitching, partial observability, amongst others.
>
> [5] Wang, R., Jansen, P., Côté, M. A., & Ammanabrolu, P. (2022). Scienceworld: Is your agent smarter than a 5th grader?. arXiv preprint arXiv:2203.07540.
> - They benchmark both online and offline RL algorithms, but focused on completing tasks related to scientific reasoning
> - No focus on interactive communication with humans/more stochastic environments, or partial observability as LMRL-Gym
>
> [6] Yao, S., Chen, H., Yang, J., & Narasimhan, K. (2022). Webshop: Towards scalable real-world web interaction with grounded language agents. Advances in Neural Information Processing Systems, 35, 20744-20757.
> - LMRL Gym has longer interactions and simulate dialog, whereas this is focuses on searching through the web
>
> [7] Yao, S., Zhao, J., Yu, D., Du, N., Shafran, I., Narasimhan, K., & Cao, Y. (2022). React: Synergizing reasoning and acting in language models. arXiv preprint arXiv:2210.03629.
> - Uses AlfWorld and Webshop which are limited, refer to [4,6]
>
> **On Offline RL**: We would like to note that most of the benchmarks for RL finetuning of LLMs are focused on online RL. Our benchmark focuses on providing an optimal testbed for both offline RL and online RL, by providing large datasets for training offline RL algorithms for LLMs, simulators for online RL training and offline evaluation, and several offline RL implementations including MC Returns, Filtered BC, and ILQL. We created the Car Dealer task to address the issues in [20] including dataset diversity. [16-19] list a series of related works in offline RL for LLMs, primarily focusing on either one task or one algorithm. Our work expands upon these works and provides a suite of both text game and dialog tasks.
>
> [8] Kumar, Aviral, et al. "When should we prefer offline reinforcement learning over behavioral cloning?." arXiv preprint arXiv:2204.05618 (2022).
>
> [9] Prudencio, Rafael Figueiredo, Marcos ROA Maximo, and Esther Luna Colombini. "A survey on offline reinforcement learning: Taxonomy, review, and open problems." IEEE Transactions on Neural Networks and Learning Systems (2023).
>
> [10] Snell, C., Kostrikov, I., Su, Y., Yang, M., & Levine, S. (2022). Offline rl for natural language generation with implicit language q learning. arXiv preprint arXiv:2206.11871.
>
> [11] Verma, S., Fu, J., Yang, M., & Levine, S. (2022). Chai: A chatbot ai for task-oriented dialogue with offline reinforcement learning. arXiv preprint arXiv:2204.08426.

---

### Official Review · Reviewer_RuRY · 2025-03-20

**Overall Recommendation:** 3

**Summary:**

The authors present 8 tasks to evaluate and build on the multi-turn capabilities of LLMs using RL. 3 tasks are interactive dialogue tasks - teaching persuasion and gather information. 5 tasks are core RL capability tasks - teaching strategic decision making, credit assignment, trajectory stitching in partially/fully observable environments (converted to text based tasks). The authors use different sized LLMs to generate seed and distill models to scale data gen. Results showcase the efficacy of value based methods, comparing against strong contemporary "baselines" such as GPT4 w/few shot and Online PPO. Though interactive dialogue tasks seem more solvable, considerable performance gap is observed between few shot GPT4 on RL capability tasks, showcasing the applicability of the benchmark.

**Claims And Evidence:**

N/A

**Essential References Not Discussed:**

N/A

**Experimental Designs Or Analyses:**

The authors design experiments to evaluate online and offline RL algorithms on core RL functionality and dialogue related tasks. They also include a strong treatment using GPT4, comparing frontier models to their approach.

Experiments and analysis using total normalized rewards provides insight into how RL algorithms can improve over simpler BC methods. BC vs ILQL shows how simple RL improves over BC. BC, ILQL vs PPO shows how newer online RL methods compare against offline ILQL.

A desired setting which is missing is using Chain-of-Thought [https://arxiv.org/abs/2201.11903] for GPT4 to reason its steps.

**Methods And Evaluation Criteria:**

Yes, the evaluation criteria using total rewards make sense for the task. Moreover, the dataset choices are sound -- making sure to include both text based and core RL capability based tasks. The supplemental material, especially section F clarifies the evaluation strategies.

**Other Comments Or Suggestions:**

N/A

**Other Strengths And Weaknesses:**

Strengths:
- Long-horizon reasoning using interactive dialogues and RL text-games is a novel approach and underpins development of complex RL long-horizon reward tasks.
- Synthetic data generation pipeline (though using smaller LLMs) serve as strong 'silver' annotations and produced high quantity of training data.

Weaknesses:
- GPT2 model is now quite old and baselines using similar sized newer models should have been reported to critically compare the performance gaps between frontier models and strong small LLMs. Newer LLMs (small) have also been trained using instruction following datasets and RL approaches. Benchmarking such LLMs along with GPT4 would have strengthened the estimation of efficacy of the dataset.
- The interactive dialogue tasks lack complexity - especially the 20Q and guess (only yes or no reply). More tasks on the lines of car dealer would strengthen the benchmark.

**Questions For Authors:**

- Most recent frontier models are also trained using Chain-of-Thought to better execute the next decision (such as ReAct framework). Did you evaluate any chain-of-thought settings with frontier models to see how well they perform?

**Relation To Broader Scientific Literature:**

Multi-turn data evaluation and generation in the context of LLMs has been a recent area of interest. This research work open sources work under-pinning the capabilities that we observe in frontier models such as long-horizon reasoning.

RL task selection and creation for LLMs to learn core RL capabilities is novel.

**Theoretical Claims:**

N/A

---

> ### Author Rebuttal · Authors · 2025-04-01
>
> We thank the reviewer for their feedback. We've addressed the main questions raised in your review by: (1) providing a justification for our choice of models (2) clarifying why we chose to have both interactive dialog tasks as well as text game tasks (3) answering your question regarding CoT for GPT-4.
>
> 1. “GPT2 model is now quite old and baselines using similar sized newer models should have been reported...”
>
> While it's true that GPT-2 is relatively older compared to more recent language models, the choice to use it in LMRL Gym was driven by its well-understood capabilities and broad usage in research for establishing baselines [3] compared to newer, similar sized models. We would like to highlight several recent works [1, 2, 4, 5] that have also used GPT2 as a baseline. Due to space limitations, we could not cite more. Our paper focuses on providing a framework that can be easily adapted to various models and the development of further algorithms for RL fine tuning for LLMs, and we hope to see further works that iterate upon other models.
>
> [1] Hicke, Y., Masand, A., Guo, W., & Gangavarapu, T. (2024). Assessing the efficacy of large language models in generating accurate teacher responses. Proceedings of the 62nd Annual Meeting of the Association for Computational Linguistics (ACL 2024). arXiv preprint arXiv:2401.12345.
> [2] Hong, J., Dragan, A. D., & Levine, S. (2024). Q-SFT: Q-learning for language models via supervised fine-tuning. Proceedings of the 42nd International Conference on Machine Learning (ICML 2024). arXiv preprint arXiv:2403.01512.
> [3] Radford, A., Wu, J., Child, R., Luan, D., Amodei, D., Sutskever, I., et al. (2019). Language models are unsupervised multitask learners. OpenAI Blog, 1(8), 9.
> 4o mini
> [4] Zhou, R., Du, S. S., & Li, B. (2024). Reflect-RL: Two-player online RL fine-tuning for LMs. Proceedings of the 62nd Annual Meeting of the Association for Computational Linguistics (ACL 2024). arXiv preprint arXiv:2401.12345.
> [5] Zhou, Y., Zanette, A., Pan, J., Levine, S., & Kumar, A. (2024). ArCHer: Training Language Model Agents via Hierarchical Multi-Turn RL. Proceedings of the 41st International Conference on Machine Learning (ICML 2024), 235, 62178–62209. arXiv preprint arXiv:2402.19446.
>
> 2. “The interactive dialogue tasks lack complexity - especially the 20Q and guess (only yes or no reply). More tasks on the lines of car dealer would strengthen the benchmark.”
>
> We would like to clarify that each of the tasks in the benchmarks serve a different purpose. Our objective in creating this benchmark is to present tasks that apply RL algorithms for multi-turn tasks in the domain of goal-directed dialogue, where agents must learn from interaction with a conversation partner. However, to enable such a large undertaking, we require tasks that can first test capabilities of RL algorithms that are essential for multi-turn dialogue, including trajectory stitching, credit assignment, and dealing with complex language. Hence, we have designed five tasks as RL Capability Tests, which are text games designed to isolate specific capabilities of RL training as shown in Figure 4. As seen, these text-games do not test all of the capabilities of RL together, which is only possible through the dialogue-based tasks. Our benchmark includes tasks that involve free-form text generation and a longer turn length. We challenge the agents in our tasks to not only follow instructions and understand the world, but plan over long trajectories, generate complex text, trajectory stitch, and resolve partial observability. For example, for the Maze and Text-Nav we test both partially observed and fully observed versions to highlight the impact of partial observability. In addition, the Text-Nav task is very similar to the Maze task, but places more emphasis on realistic text.
>
> Lastly, the dialogue tasks have been designed with increasing levels of difficulty, with twenty questions testing the ability of RL algorithms to perform information gathering, guess my city testing the ability to ask questions beyond just yes/no and with free form feedback, and the Car Dealer task to test more strategic decision making and persuasive capabilities of RL algorithms for LLMs. As shown, some tasks aim to test specific RL properties without the complexities of realistic language, while others focus on complex language. We wanted our tasks to cover a range of RL capabilities to isolate issues with algorithms, as a group of complicated and difficult tasks may not provide such understanding and insight.
>
> 3. “Did you evaluate any chain-of-thought settings with frontier models?”
>
> We appreciate the suggestion to evaluate GPT-4 and other models using CoT reasoning. While CoT has demonstrated strong performance in reasoning-based tasks, our focus was on evaluating baseline RL capabilities without extensive prompt engineering. and we wanted a fair comparison across all settings, including human evaluation and testing on our algorithms.

---

### Decision · Program_Chairs · 2025-05-01

**Decision:**

Accept (poster)

**Comment:**

The authors present LMRL Gym, a benchmark (and environment) for evaluating multi-turn RL with LLM agents in offline and online settings. Specifically, they provide three (progressively challenging) interactive dialogue tasks and five RL capability tasks (teaching strategic decision making, credit assignment, POMDP optimization, trajectory stitching, and complex language generation). -- addressing a gap for RL evaluation within LLM settings. The LMRL-Gym is evaluated on multiple RL algorithms including behavior cloning, offline value-based RL, and online PPO variants -- along with a few-shot GPT-4 implementation for comparison. Several interesting observations regarding these experiments are presented in section 6, demonstrating that the LMRL-Gym is effective at testing scientific hypotheses empirically.

Strengths of this work identified by reviewers include:
- This work does identify a gap in RL using LLMs by providing natural language tasks that require longer horizon sequential decision making with progressively challenging targeted tasks to demonstrate specific capabilities. The resulting benchmark is non-trivial and would likely be challenging for state-of-the-art algorithms and can detect methodological improvements.
- The approach for building LMRL-Gym is clear when including the Appendix materials.
- There will be a released environment for other researchers to build on (and is believed to be useful in advancing the field).

Limitations of this work identified by reviewers includes:
- While related works have relied on GPT-2, there are newer 'smaller' LLMs that have predictable and well-understood behavior that may produce a stronger benchmark. This should at least be discussed within the paper (i.e., how LMRL-Gym would continue to improve with stronger LLMs).
- The choice of baseline models is limited. While this is a somewhat an open-ended request (i.e., one could always ask for more models), it would be helpful to include more models for which implementations are available and/or clarify how easily more methods can be added.
- A more thorough comparison with existing related multi-turn benchmarks. This was largely addressed in rebuttal and should be added to the main text and/or appendices as space permits.

Overall, I believe the reviewers recognized the importance of the contribution -- although I believe there was some confusion regarding the significance and relation to existing benchmarks (which was sufficiently addressed in rebuttal in my opinion). I believe other concerns were a clear path regarding how to continue improving the benchmarks in LMRL-Gym (e.g., through stronger LLMs) and the strength of the provided baselines (and utility of the corresponding conclusions). If these components were stronger, I believe the reviewers would be more confident that LMRL-Gym will have a longer horizon for utility and be able to confidently detect improvements and empirically validate hypotheses (i.e., Section 6 should be written with more rigor). That being said, I believe that this is a useful and potentially impactful work even with these limitations.